# Asymmetric gradient orbital interaction of hetero-diatomic active sites for promoting C − C coupling

Jin Ming Wang[1], Qin Yao Zhu[1], Jeong Heon Lee[1], Tae Gyun Woo[1], Yue Xing Zhang ®[2], Woo-Dong Jang ®[1] & Tae Kyu Kim ®[1] ✉

Diatomic-site catalysts (DACs) garner tremendous attention for selective $CO_2$ photoreduction, especially in the thermodynamical and kinetical mechanism of $CO_2$ to $C_{2+}$ products. Herein, we first engineer a novel Zn-porphyrin/RuCu-pincer complex DAC (ZnPor-RuCuDAC). The heteronuclear ZnPor-RuCuDAC exhibits the best acetate selectivity (95.1%), while the homoatomic counterparts (ZnPor-$Ru_2$DAC and ZnPor-$Cu_2$DAC) present the best CO selectivity. In-situ spectroscopic measurements reveal that the heteronuclear Ru–Cu sites easily appear $C_1$ intermediate coupling. The in-depth analyses confirm that due to the strong gradient orbital coupling of Ru$4d$–Cu$3d$ resonance, two formed ˙CO intermediates of Ru–Cu heteroatom show a significantly weaker electrostatic repulsion for an asymmetric charge distribution, which result from a side-to-side absorption and narrow dihedral angle distortion. Moreover, the strongly overlapped Ru/Cu-$d$ and CO molecular orbitals split into bonding and antibonding orbitals easily, resulting in decreasing energy splitting levels of $C_1$ intermediates. These results collectively augment the collision probability of the two ˙CO intermediates on heteronuclear DACs. This work first provides a crucial perspective on the symmetry-forbidden coupling mechanism of $C_1$ intermediates on diatomic sites.

Photo-driven $CO_2$ neutralization into chemical fuels is considered a promising strategy for energy storage and greenhouse-effect alleviation[1–4]. Presently, the most common products of photocatalytic $CO_2$ reduction reaction ($CO_2$RR) are $C_1$ fuels such as carbon monoxide (CO)[5], methane ($CH_4$)[6,7], and formate (HCOOH)[8], even though tremendous descriptors have been proposed for regulating product selectivity of $CO_2$RR by controlling the catalytic centers, chromophores, and charge separation efficiency[9–12]. Among various photocatalysts, intraskeletal photogate molecular device (PMD)-derived catalysts are considered ideal protocols that integrate chromophoric photosensitizers, atomic metal catalytic sites, and electron-transfer mediators to form a full-component molecular photogate system[5,13,14]. However, because of the high energy barrier and sluggish multiple-electron kinetics of the C−C coupling process, the selective

$C_{2+}$ (ethylene, ethanol, acetate, etc.) yield is noticeably low on PMD catalysts. In particular, the C−C coupling is impeded during the $C_2$ formation because the atoms of adjacent $C_1$ intermediates exhibit almost identical charge distributions, which practically lead to a strong electrostatic repulsive force that induces dipole−dipole interactions between the intermediate molecules[15–17]. Therefore, for developing efficient photocatalysts to stimulate the C−C coupling process as well as achieve a high $C_{2+}$ selectivity, optimizing the charge distribution and spatial structure of the $C_1$ intermediates (especially for ˙CO intermediates) are essential steps.

Recently, diatomic-site catalysts (DAC) have been proposed to enhance the $CO_2$RR selectivity by endowing the excited reaction sites with different charge enrichment[18–21]. In this process, two neighboring atomic metal species are utilized because of their synergistic actions

[1]Department of Chemistry, Yonsei University, Seoul 03722, Republic of Korea. [2]College of Chemistry and Chemical Engineering, Dezhou University, Dezhou 253023, China. ✉e-mail: tkkim@yonsei.ac.kr

and complementary functionalities. Particularly, the absorption behavior and binding strength of the intermediates can be tailored at the DACs by tuning the electronic reciprocal action of the dual-atom sites via partial oxidation/reduction[8,22–24], doping[25], or atom vacant interventions[26–28]. For example, a Cu-doped hybrid photocatalyst can reduce $CO_2$ to $C_2H_4$ with 32.9% selectivity;[29] the unique $Cu^+$ surface layer is the active site, where the in situ generated $^*CO$ can be anchored, and further C–C coupling can be performed to form $C_2H_4$. Further, an In–Cu dual-metal photocatalyst can support the photocatalytic $CO_2$ conversion to ethanol with a high selectivity of 92%[30]. The In–Cu dual-metal sites promote the adsorption of $^*CO$ intermediates and lower the energy barrier of the C–C coupling. Theoretically, the intermediate atoms absorbed on the symmetrical homonuclear DAC sites show an equal electron scattering facilely[15,31], which impedes the subsequent C–C coupling pathways (Fig. 1a). Comparatively, because of the distinct electronegativity and gradient orbital coupling at the heteroatoms, the intermediates on heteronuclear DACs possibly exhibit asymmetric charge distributions, which can suppress the repulsive molecular interactions of the formed intermediates and augment the collision probability of adjacent intermediates to couple and form $C_{2+}$ products. A promising strategy is to merge the closely-lying PMDs into covalent organic frameworks (COFs) with diatomic sites, facilitating an efficient energy transfer between the bonded photosensitizers and dual-atom centers at the COF compartments. Therefore, an in-depth insight into the selectivity mechanism of heteronuclear and homonuclear diatomic sites is necessary to reveal the unique advantages of DACs, aiding in the rational design of PMD-derived photocatalysts with high catalytic performances and multifunctionalities via structural and spatial isomerization.

Inspired by the above considerations, we synthesized AB-stacked heteronuclear dual-atom-site COFs by self-assembling Zn-porphyrin and Ru/Cu-pincer complexes for photocatalytic $CO_2$RR (Supplementary Fig. 1, ZnPor-RuCuDAC). Further, we fabricated homonuclear counterparts (ZnPor-$Ru_2$DAC and ZnPor-$Cu_2$DAC) for comparison and exploring the diatomic-type effect on product selectivity of $CO_2$RR. Because of ZnPor-RuCuDAC's high crystallinity, one-dimensional (1D) pore channels and individual building skeletons was observed. The experimental results and theoretical calculations confirmed that an efficient intraskeletal electron transfer can occur from Zn to the diatomic sites under light irradiation. A detailed electronic structure analysis of the crucial $^*CO$ intermediates revealed that the heteroatomic Ru–Cu pair serving as a catalytic center showed a strong Ru4$d$–Cu3$d$ gradient orbital coupling. This resulted in an asymmetric charge distribution of the two formed $^*CO$ intermediates, which parallelly emerged on the Ru–Cu atoms in a side-by-side configuration and weakened the energy barrier of the C–C coupling process. The findings of this study present critical insights into the rational designing, working mechanism, and applications of hetero-DACs.

## Results and discussion
### Synthesis and microstructural analyses
The hydrazone-linked dual-atom-site COFs were prepared by condensing 5,10,15,20-Tetrakis(3,5-dibenzaldehyde) porphyrin Zn(II) (ZnPor)[32,33] and 2,6-bis(5-amino-1H-benzimidazol-2-yl)pyridine Ruthenium or Copper(II) ($RuN_3$ or $CuN_3$)[34,35] monomers (Supplementary Fig. 1). The chemical structures of these COFs were analyzed by Fourier transform infrared (FTIR) spectroscopy and solid-state nuclear magnetic resonance (NMR). The FTIR spectrum of ZnPor (Supplementary Fig. 2) shows the characteristic stretching vibration bands of C = O and C-H, presenting in the aldehyde group[36,37] at 2852/2926 $cm^{-1}$. The spectra of $RuN_3$ and $CuN_3$ monomers exhibit two peaks at 3205 and 3336 $cm^{-1}$, which originate from the stretching vibrations of amino groups[34], and the peak at 615 $cm^{-1}$ can be assigned to the vibration of metal–Cl (Ru–Cl or Cu–Cl) in the monomer. In these COFs, the

stretching vibrations of formyl moiety and the amino group almost disappear, and a new FTIR band (-1607 $cm^{-1}$) associated with the stretching vibration of –C=N– bond appears (Supplementary Fig. 3), indicating the formation of hydrazone linkages[38,39]. Further, the disappeared FTIR peak corresponding to the metal–Cl bond can be ascribed to the chloride ion removal of $RuN_3$ and $CuN_3$ ligands in these COFs' scaffolds. The $^{13}C$ NMR spectra (Supplementary Fig. 4) of all the COFs exhibit a characteristic signal at -162.3 ppm[36], thereby revalidating the formation of hydrazone bonds. The results of elemental analysis by inductively coupled plasma-atomic emission spectrometry (Supplementary Table 1) indicate that the experimental weight percentage of elemental C, H, N, Zn, Ru, and Cu are close to their respective theoretical compositions[40], also suggesting their relatively high purity.

To elucidate the structures of these COFs and unit cell parameters, four types of possible 2D structures were generated for ZnPor-RuCuDAC (Supplementary Fig. 5 and Supplementary Tables 2–4): AA-eclipsed, AB-staggered, slipped ABC-staggered and ABCD- staggered stacking models[41]. Also, the density functional tight binding method with the Lennard–Jones function was utilized to evaluate crystal stacking energies of ZnPor-RuCuDAC quantitatively (Supplementary Table 5). Evidently, the ABCD-staggered model exhibits the highest total per-layer crystal stacking energy (193.95 kcal $mol^{-1}$) as compared to that of the others, indicating that ABCD-staggered stacking is more favorable than the other four isomeric structures. The experimental PXRD profiles of ZnPor-RuCuDAC (Fig. 1b), ZnPor-$Ru_2$DAC (Fig. 1c), and ZnPor-$Cu_2$DAC (Fig. 1d) exhibit three main diffraction peaks at 3.06°, 4.42°, and 9.57°, matching well with the simulated diffraction pattern of ABCD-staggered stacking model evidenced by the negligible difference obtained in the Pawley refinement results. The details of crystal structures associated with different stacking models, including the unit cell parameters, are available in the supplementary information (Supplementary Table 5). The porosity of these COFs was evaluated from the nitrogen sorption isotherms (Supplementary Figs. 6–8), and the measured pore sizes were 1.66 (ZnPor-RuCuDAC COF), 1.68 (ZnPor-$Ru_2$DAC COF), and 1.65 (ZnPor-$Cu_2$DAC COF) nm, further confirming the ABCD-staggered stacking geometry of these COFs due to the complete agreement between experimental and theoretical pore sizes. As contrasted to the original PXRD pattern, ZnPor-RuCuDAC (Supplementary Fig. 9a) do not vary in the peak position and intensity under different solvents obviously, indicating highly chemical-stable crystallinity in acid/alkaline conditions. From thermogravimetric analysis (Supplementary Fig. 9b–d), all three COFs are thermally stable up to 368 °C.

Scanning transmission electron microscopy (STEM) was performed to examine the ZnPor-RuCuDAC microstructures. The high-resolution aberration-corrected transmission electron microscopy (AC-TEM) images (Supplementary Fig. 10a, b) exhibit transparent nanosheets with lateral dimensions. The elemental analysis results of the ZnPor-RuCuDAC crystal (Supplementary Fig. 10c–f) show a uniform distribution of the Zn, Ru, and Cu elements on the scaffolds. From aberration-corrected bright-field STEM (AC-BF-STEM) of ZnPor-RuCuDAC COF, we clearly observe a periodic square pore skeleton with well-defined building units (Fig. 1e) for orderly stacked structure and high crystallinity[42,43]. The corresponding color-coded contour map (Fig. 1f) shows distinct monomer molecular compartments and grid-shaped 1D pore channels[44]. Interestingly, aberration-corrected annular dark-field STEM (AC-ADF-STEM), with an emerging differential phase contrast (DPC) technique at an extremely low beam current, was applied to generate an integrated DPC image over the atom distribution in the COF backbone[6]. The parameters of individual building compartments are measured with $a = b = 1.67 \pm 0.1$ nm ($\alpha = 90 \pm 0.3°$) in ZnPor-RuCuDAC COF, in good agreement with the ABCD-staggered pore size of the simulated structure (Fig. 1g

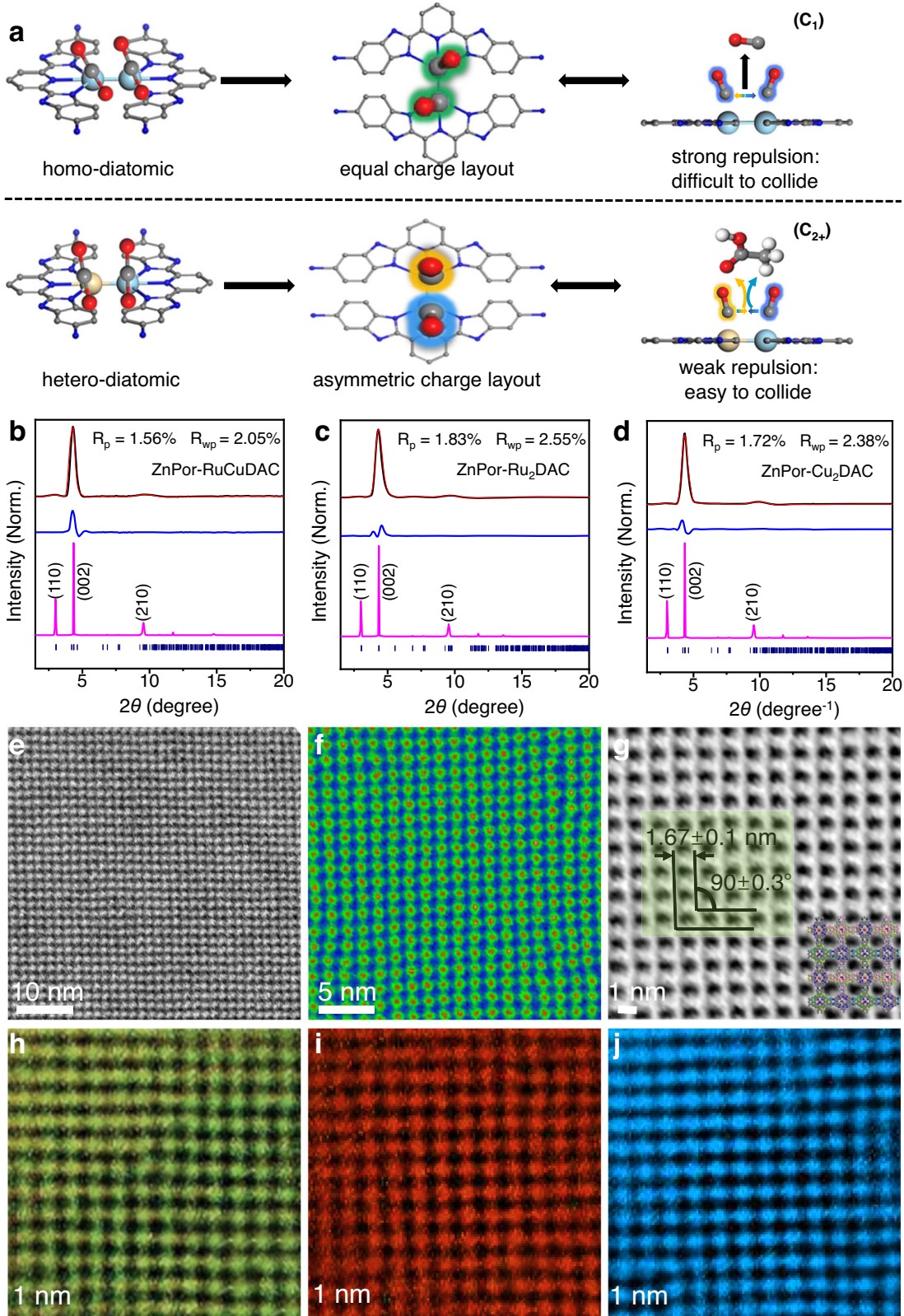

**Fig. 1 | Selective reaction mechanism and microstructural characterization.** **a** Expected photocatalytic CO$_2$ reduction mechanism of C$_1$ and C$_{2+}$ product formation using homo− and hetero-diatomic catalytic sites, respectively. **b**–**d** Pawley-refined PXRD of **b** ZnPor-RuCuDAC, **c** ZnPor-Ru$_2$DAC, and **d** ZnPor-Cu$_2$DAC. **e**–**j** STEM analyses of ZnPor-RuCuDAC. **e** Filtered AC-BF-STEM images and **f** corresponding color-coded pore channel map. **g** DPC image using AC-ADF-STEM and **h**–**j** corresponding 3D electron diffraction tomography of DPC images. All microstructural analyses confirm the ABCD-staggered stacking structure of ZnPor-RuCuDAC COF.

and Supplementary Fig. 5f, g). The 3D electron diffraction tomography of DPC images (Fig. 1h–j) unveil the spatial distances among adjacent Zn, Cu, and Ru are *ca.* 1.67 nm over the ZnPor-RuCuDAC substrate, where the Zn and Ru/Cu atoms show a 90° phase difference on the spatial location (Supplementary Fig. 11). These results confirm that the COFs possibly present ABCD-staggered stacking structure.

X-ray absorption near edge structure (XANES) and extended X-ray absorption fine structure (EXAFS) measurements were carried out to evaluate the local coordination environment and electronic structures of these diatomic-site catalysts. As shown in Fig. 2a–c and Supplementary Table 6, within COF matrices, the pre-edge energies of Zn K-edge XANES (Fig. 2a) are shifted toward higher photon energies compared to those for the ZnPor monomer, which exhibits a $3d^{10}$ ($S = 0$) oxidation peak corresponding to $Zn^{II}$ phthalocyanine (ZnPc)[40]. The COFs' Ru pre-edge peak (Fig. 2b) shows a much more negative edge position than the $RuN_3$ sample, which exhibits a $4d^6$ ($S = 0$) electronic state peak at a photon energy ranging between the energies of the Ru foil and $RuCl_3$[14]. Similarly, all COFs' Cu K-edge absorption

peak (Fig. 2c) also shows a negative shift compared to that of $CuN_3$ close to $Cu^{II}$ phthalocyanine (CuPc) with $3d^9$ ($S = 1/2$) spin ground state[29]. The shifting phenomenon is further shown by the first derivatives of the spectra (bottom layer in Fig. 2a–c). This observation implies that the excited-state intramolecular energy transfer reduces the electron density on the ZnPor unit and increases that on the $RuN_3$/$CuN_3$ cores under X-ray illumination[40,45]. According to the Fourier-transformed radial distribution functions of the EXAFS spectra of ZnPor and these COFs (Fig. 2d and Supplementary Table 7), the first-shell Zn K-edge peak (1.44 Å) is close to the ZnPc peak (1.45 Å), coordinating with the central 4.0 N ($Zn-N_4$) shell. By contrast, no additional Zn–Zn atomic scattering, as referred to Zn foil (at 2.66 Å), is detected because of the single-atom site of the Zn species, with 4.1 N, 3.9 N, and 4.0 N atom at the ZnPor-RuCuDAC, ZnPor-$Ru_2$DAC, and ZnPor-$Cu_2$DAC COFs. Moreover, compared to the Ru foil and $RuCl_3$ reference (Fig. 2e and Supplementary Table 8), the R space Ru K-edge spectra of $RuN_3$ show coordination peaks of Ru–N (1.43 Å) and Ru–Cl (1.99 Å), but no Ru–Ru (2.76 Å) peaks. Similarly, in the case of $CuN_3$, the R space Fourier transformed Cu K-edge spectrum (Fig. 2f and

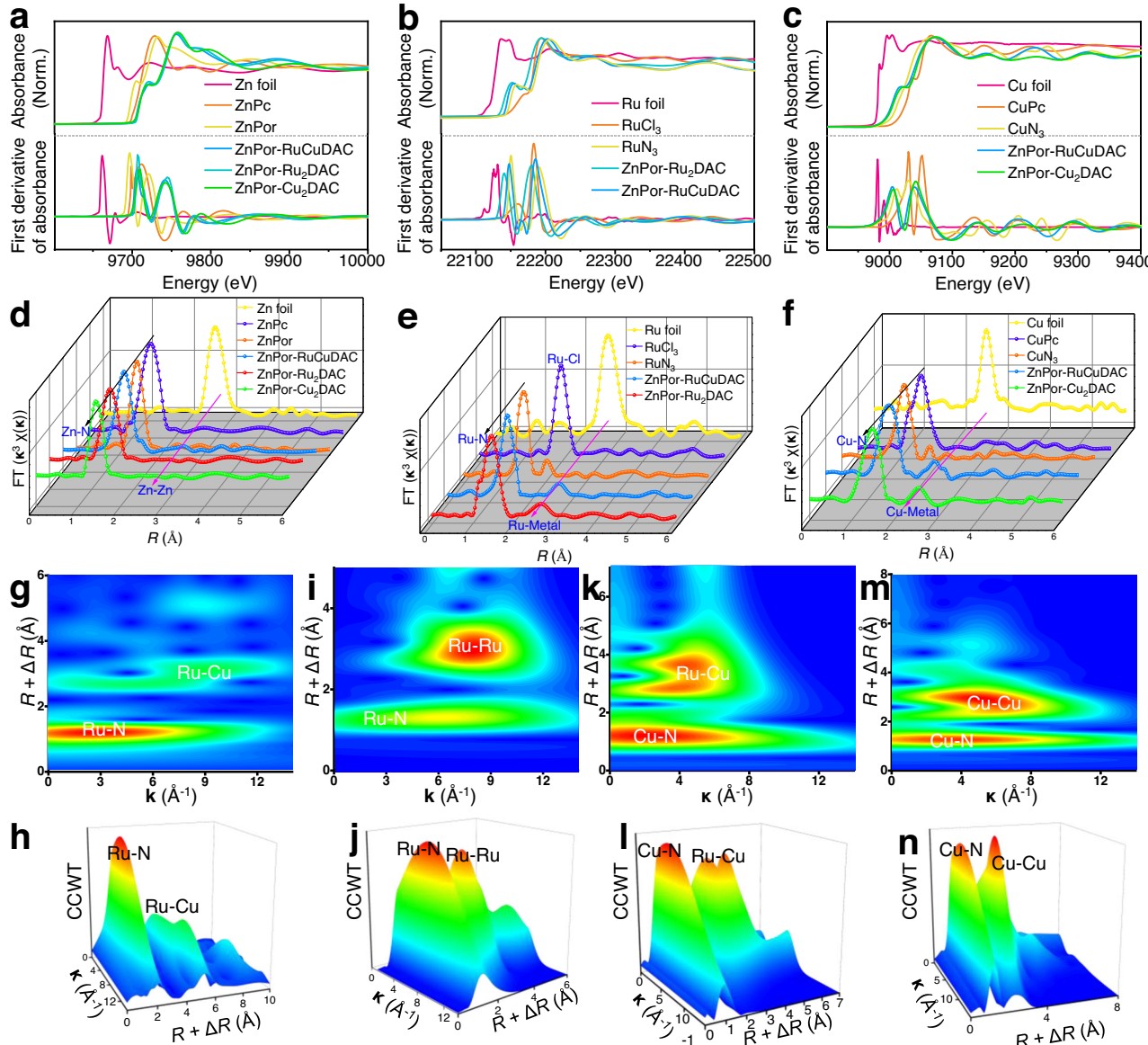

**Fig. 2 | X-ray absorption spectroscopy (XAS) measurements of diatomic COFs.** **a** Zn, **b** Ru, and **c** Cu K-edge XANES spectra and their first derivatives. **d** Zn, **e** Ru, and **f** Cu K-edge $k^3$-weighted EXAFS spectra. **g**–**n**. Continuous Cauchy wavelet

transforms (CCWT) of $k^3$-weighted EXAFS profiles of Ru K-edge of **g**, **h** ZnPor-RuCuDAC and **i**, **j** ZnPor-$Ru_2$DAC and Cu K-edge of **k**, **l** ZnPor-RuCuDAC and **m**, **n** ZnPor-$Cu_2$DAC.

Supplementary Table 9) exhibits peaks at 1.47 and 2.07 Å, corresponding to the Cu–N and Cu–Cl shell scatterers, respectively, and no additional Cu–Cu interaction referring to the Cu foil (at 2.61 Å) is observed. However, the Ru–Cl and/or Cu–Cl signals disappear in the spectra of the ZnPor-RuCuDAC, ZnPor-Ru$_2$DAC, and ZnPor-Cu$_2$DAC backbones because of the Cl$^-$ disassociation from the Ru and/or Cu sites during the condensation of the ZnPor, RuN$_3$, and CuN$_3$ monomers, indicating no Cl impurity on the COFs in consistent with element analysis.

Comparatively, the predominant peak of Ru centers of ZnPor-RuCuDAC and ZnPor-Ru$_2$DAC (Fig. 2e) exhibit 3.1 N and 6.0 N coordination of Ru–N path scattering, which is almost like that of RuN$_3$ for nearly identical atom coordination environment. However, new peaks are observed at 2.70 (ZnPor-RuCuDAC) and 2.68 Å (ZnPor-Ru$_2$DAC)[15], suggesting the presence of Ru–metal (Ru–Cu and Ru–Ru) diatomic configuration with 1.0 atom coordination. On the other hand, if we exclude the 3.1 N and 5.9 N Cu–N (1.47 Å) coordination in the ZnPor-RuCuDAC and ZnPor-Cu$_2$DAC (Fig. 2f), then the emerging Cu–metal peaks (1.0 atom coordination) that originate from the Cu foil reference indicate the formation of Cu–metal (Cu–Ru, Cu–Cu) dual-atom sites[17]. According to the EXAFS fitting results (Supplementary Tables 8, 9), the Ru and/or Cu centers emerge with a coordination number of 1.0 for the Ru–Cu (ZnPor-RuCuDAC), Ru–Ru (ZnPor-Ru$_2$DAC), and Cu–Cu (ZnPor-Cu$_2$DAC) bonds in addition to the coordination with the central N atoms. Wavelet transformation of the $k^3$-weighted EXAFS was also conducted to identify the metal–N and metal–metal paths. The Zn centers in these three COFs merely show intensity maxima ($R + \Delta R = 1.43$ Å) for the Zn–N path (Supplementary Fig. 12). However, the Ru centers of ZnPor-RuCuDAC (Fig. 2g, h) and ZnPor-Ru$_2$DAC (Fig. 2i, j) show maximum intensity for the Ru–N and Ru–metal paths, respectively, compared to RuN$_3$ (Supplementary Fig. 13a). Further, both ZnPor-RuCuDAC (Fig. 2k, l) and ZnPor-Cu$_2$DAC (Fig. 2m, n) exhibit Cu–N and Cu–metal shells, unlike CuN$_3$ reference (Supplementary Fig. 13b). Compared to ZnPor and RuN$_3$/CuN$_3$ monomers, all COFs present Zn(II) $2p_{3/2}/2p_{1/2}$ binding energy (BE) peaks, ascribable to Zn–N bond (Supplementary Fig. 14), but emerge new BE peaks of Ru–Cu (ZnPor-RuCuDAC), Ru–Ru (ZnPor-Ru$_2$DAC), and Cu–Cu (ZnPor-Cu$_2$DAC) diatomic pairs besides Ru–N and/or Cu–N bond[15]. These results collectively indicate that compared to the single-atom Zn distribution, ZnPor-RuCuDAC mainly exists Ru–Cu diatomic pairs via diatomic coordination assemblies of metal–N$_3$ connected by Ru–Cu bonding bridges even though few homonuclear appear possibly, while ZnPor-Ru$_2$DAC and ZnPor-Cu$_2$DAC are anchored the Ru–Ru and Cu–Cu sites.

To discover the diatomic-type effect on the electronic band structure, synchrotron radiation photoemission spectroscopy (SRPES) was conducted on these COFs. The SRPES results indicate that their valence band (VB, vs. NHE) potentials are 1.05 (ZnPor-RuCuDAC), 0.95 (ZnPor-Ru$_2$DAC), and 0.93 (ZnPor-Cu$_2$DAC) eV (Supplementary Fig. 15a–c). Based on the formula ($\varphi = h\nu - E_{cutoff}$, where $\varphi$ is the workfunction, h$\nu$ is the excitation energy, and $E_{cutoff}$ is the cut-off energy of the secondary electron), we can conclude that the workfunction of ZnPor-RuCuDAC (4.34 eV) is lower than those of ZnPor-Ru$_2$DAC (5.03 eV) and ZnPor-Cu$_2$DAC (5.17 eV) (Supplementary Fig. 15d–f). The ultraviolet-visible diffuse reflection spectra (DRS, Supplementary Fig. 16a) of all COFs show intense absorptions in the range of 300–817 nm, indicating their excellent light harvesting characteristics[40,45]. The Kubelka–Munk method (Supplementary Fig. 16b–d) was employed to determine the optical bandgap energies ($E_g$), which were found to be 1.75 (ZnPor-RuCuDAC), 1.63 (ZnPor-Ru$_2$DAC), and 1.66 (ZnPor-Cu$_2$DAC) eV. Therefore, the relative energy band structures of these COFs (Supplementary Table 10) indicate that the conductive band potentials are appropriate for the formation of typical photocatalytic products (Supplementary Fig. 17a and Table 11), theoretically preferable for CO$_2$RR.

## Analyses of CO$_2$ photoreduction activity

CO$_2$ reductions over various monomers and diatomic COFs were conducted using a liquid-phase photoreaction system, with triethanolamine assistance as a sacrificial agent upon oxidized process (Supplementary Fig. 18). The ZnPor monomer does not exhibit any CO$_2$RR activity (Fig. 3a). However, the production rate of H$_2$, CO, and formate is higher for CuN$_3$ than for RuN$_3$, indicating that the chelated copper centers can act as more efficient active sites for CO$_2$RR. To reveal the effect of the diatomic types on the product generation, a series of photocatalytic CO$_2$RRs were conducted with various Ru:Cu molar ratios from 0:1 to 1:0. Compared with the monomers, the diatomic COFs interestingly exhibit acetate production (Fig. 3b), which gradually varies (in a parabolic trend) with the increasing Ru:Cu molar ratio from 0:1 to 1:0. Concurrently, the H$_2$ generation rate decreases as Ru:Cu molar ratio increase (Supplementary Fig. 19), whereas both the CO and formate evolution rates first significantly reduce and then show a slight improvement (Fig. 3c and Supplementary Fig. 20). This result suggests that the diatomic-site types in the COFs can significantly affect the product species. Among these photocatalysts, the Ru:Cu = 0.5:0.5 COF (for simplicity, denoted as ZnPor-RuCuDAC) delivers the highest acetate production rate (400.5 μmol g$^{-1}$ h$^{-1}$) as well as shows the lowest formate and CO photoactivities. The Ru:Cu = 0:1 COF (denoted as ZnPor-Cu$_2$DAC) presents a CO evolution rate of 321.9 μmol g$^{-1}$ h$^{-1}$ (Fig. 3c), which is 6.1 times higher than that of the Ru:Cu = 1:0 COF (denoted as ZnPor-Ru$_2$DAC). Notably, among the diatomic COFs with different Ru:Cu molar ratios (Fig. 3d, e), ZnPor-RuCuDAC exhibits the highest acetate selectivity of 95.1%, followed by ZnPor-Cu$_2$DAC (36.9%) and then ZnPor-Ru$_2$DAC (21.6 %). Further, ZnPor-Cu$_2$DAC presents the most superior CO selectivity. These variable trends on photoactivity and selectivity of CO$_2$RR with different Ru:Cu molar ratios can be attributed to the active-site-type variations in the COF skeleton, which can influence the absorption behavior and reaction pathways of intermediates.

In addition to the high acetate selectivity of ZnPor-RuCuDAC, a stable photoactivity was also retained (Fig. 3f and Supplementary Figs. 21–23), and no obvious decay was observed during four consecutive runs. Moreover, the diatomic catalyst and its recovered composite present quite similar DRS (Supplementary Fig. 24a), PXRD (Supplementary Fig. 24b) and XPS spectra (Supplementary Fig. 25), suggesting that crystallinity, element composition, and oxidation state are still maintained after 40 h reaction. Additionally, no products were detected without the photocatalyst, CO$_2$, or light irradiation (Supplementary Fig. 26). Therefore, it is confirmed that photoinduced electrons trigger the CO$_2$RR and initiate the C–C coupling, leading to acetate formation. To further confirm the carbon source of photogenerated products, $^{13}$CO$_2$ isotope tracing was performed during the reaction, and the corresponding mass spectra (Fig. 3g–i) and total ion chromatograms (Supplementary Figs. 27 and 28) were analyzed. The mass spectrum of acetate fraction shows a set of peaks at m/z from 45 to 62, consistent with that of the molecular ions of $^{13}$CH$_3$$^{13}$COOH$^+$ (m/z = 62) and fragments ions of $^{13}$CH$_3$$^{13}$COO$^+$ (m/z = 61), $^{13}$COOH (m/z = 46), and $^{13}$CH$_3$$^{13}$CO$^+$ (m/z = 45). This result indicates that the acetate fraction exhibits an isotope-induced mass shift effect of ($M + 2$) or ($M + 1$), referring to the non-isotope labeled CH$_3$COOH. It is worth noting that the largest peak in the mass spectrum belongs to the $^{13}$CH$_3$$^{13}$COO$^+$ due to the easier dissociation of acetate molecules. The isotope-labeled H$^{13}$COOH product has H$^{13}$COOH$^+$ (m/z = 47) molecular ions and fragments ions ($^{13}$COOH: m/z = 46, $^{13}$COO$^+$: m/z = 45, H$^{13}$CO$^+$: m/z = 30, and $^{13}$CO$^+$: m/z = 29), which show an ($M + 1$) mass shift effect compared to the non-isotope labeled HCOOH. In addition, the molecular and fragment ions obtained from $^{13}$CO ($^{13}$CO m/z = 29, $^{13}$C$^+$ m/z = 13, $^{16}$O$^+$ m/z = 16) exhibit a mass shift effect ($M + 1$) compared with the non-isotope labeled CO. These results confirm that the obtained products (acetate, formate, and CO) are indeed derived from the CO$_2$RR rather than the other carbon sources.

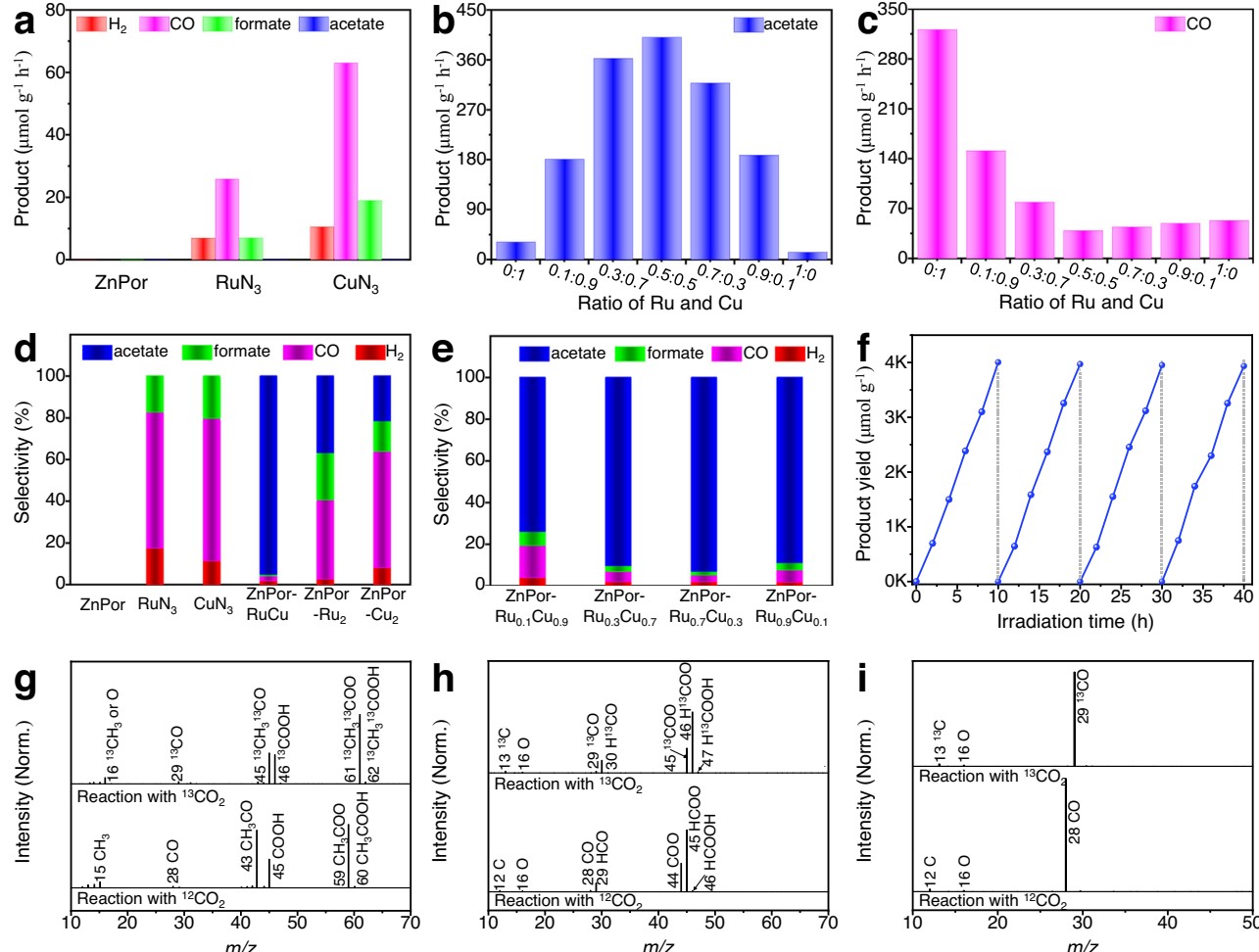

**Fig. 3 | Photocatalytic CO₂ reduction assessments. a** Evolution rates of H₂, CO, formate, and acetate products from monomers (ZnPor, RuN₃, and CuN₃). **b**, **c** Evolution rates of **b** acetate and **c** CO from ZnPor-RuCuDAC with different Ru:Cu molar ratios. **d**, **e** Product selectivity (%) of photocatalytic CO₂ reduction products from **d** monomers and diatomic COFs and **e** ZnPor-RuCuDAC with different Ru:Cu molar ratios. **f** Cycling tests of CO₂ photoreduction to acetate of ZnPor-RuCuDAC. **g**–**i** ¹³C Isotope labeling mass spectra. ¹³C and ¹²C mass spectra of **g** acetate, **h** formate, and **i** CO were obtained from the photocatalytic ¹³CO₂ and ¹²CO₂ reductions over ZnPor-RuCuDAC.

## Mechanism of photoactivity enhancement

Quasi in situ Raman spectroscopic analyses were carried out to identify possible reactive sites on diatomic COFs under the same 53 °C temperature as the photocatalytic CO₂ reduction experiment. After the ZnPor-RuCuDAC surface was initially cleaned using pure Ar gas in the dark condition, pure CO₂ was purged into the chamber, and the quasi in situ Raman signals were intermittently recorded. As shown in Fig. 4a, the peak positions of both the Ru−Cu stretching (~226 cm⁻¹) and bending (~252 cm⁻¹) vibrations are consistent with the simulated one (Supplementary Fig. 29), which gradually shift to higher wavenumbers during CO₂ adsorption and show no change at 60 min for the saturated CO₂ adsorption. The observed 3 nm blue-shifting indicates that the orbital coupling between M-$d$ (M represents Ru or Cu-metal atom) and C-$2p$ orbitals promote electron cloud migration from the metal sites to the CO₂ orbital, leading to CO₂ molecular activation. Under illumination, the enriched photoelectron cloud of the Ru-Cu sites further moves to the absorbed CO₂ molecular orbitals for the photoreduction of CO₂ and reaction intermediates, resulting in Ru−Cu bonds with changeable strengths that again promote Ru−Cu vibration peaks shifting to higher frequencies (Fig. 4b)[46]. Furthermore, as seen in the variation of Raman peak position obtained from different measurements (Fig. 4c, blue rectangle of the light-on and light-off spectra), the blue-shifted Raman vibration band (~202–216 cm⁻¹) at Ru–Cu sites

emerges during 0–90 min of light irradiation in reference to the ground-state spectrum. This new band may be attributed to the deformation vibration of Ru–Cu sites, containing the absorbed C–C coupling intermediates during CO₂RR[30,46,47]. After measurement (Fig. 4b, c), the deformation vibration band disappears, and the Ru−Cu stretching and bending vibration peaks return to their original states within 30 min light-off, suggesting favorable reversibility of Ru−Cu diatomic site.

To comprehensively reveal the charge-transfer mechanism between the ZnPor unit and dual-atom active sites, the XAS of metal L-edge was studied since it can directly determine whether the illumination causes a change in the electronic structure of metal centers at the COFs[15]. Under different irradiation times, the Zn L₃-edge of ZnPor-RuCuDAC shifts toward higher photon energies (Supplementary Fig. 30), whereas the Ru and Cu L₃-edges move to the lower photon-energy side (Supplementary Fig. 31). The picosecond transient absorption (ps-TA) spectroscopy of ZnPor-RuCuDAC (Supplementary Fig. 32) exhibit the broader and more intensive transient absorption band at 552–736 nm than the metal-deficient COF counterparts (ZnPor-N₃ COF and H₂Por-RuCuDAC), ascribable to the charge transfer between ZnPor (Por) cores and Ru–Cu diatomic sites. The time-resolved photoluminescence (TRPL) spectrum of ZnPor-RuCuDAC (Supplementary Fig. 33 and Table 12) shows the fastest decay lifetime

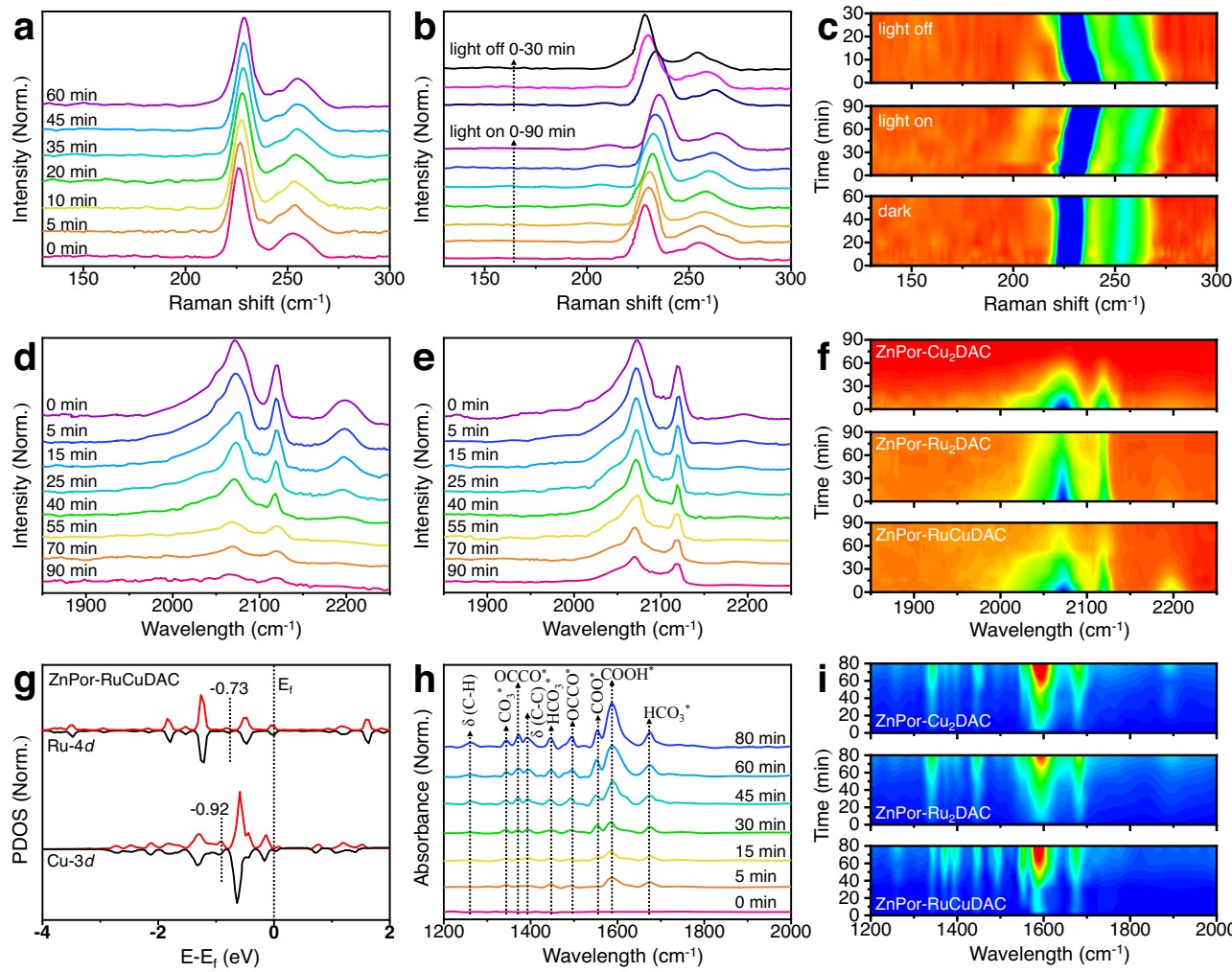

**Fig. 4 | The in situ spectroscopic analyses of photocatalytic $CO_2$ reduction of COFs. a–c** Quasi in situ Raman spectra of ZnPor-RuCuDAC obtained under **a** dark, **b** light-on, and light-off conditions, and **c** the corresponding 2D contour plots. **d–f** In situ DRIFTS profiles of CO desorption on **d** ZnPor-RuCuDAC and **e** ZnPor-Ru$_2$DAC at different reaction times, and **f** the corresponding 2D contour plots of the three COFs. **g** Ru 4$d$ and Cu 3$d$ partial DOS on ZnPor-RuCuDAC. **h** The in situ DRIFTS profile of ZnPor-RuCuDAC during $CO_2RR$ and **i** the corresponding 2D contour plots of the three COFs.

than ZnPor-$N_3$ COF and $H_2$Por-RuCuDAC, especially with the 49.6% reduced $\tau_2$ decay lifetime and the shortest average decay lifetime (1.53 ns). These observations suggest that the Ru–Cu centers can accept the photogenerated electrons of ZnPor units and act as the dual-atom active sites (Supplementary Fig. 17b)[40,42], leading to the formation of low-valence Ru–Cu species during the illumination.

Photocatalytic $CO_2RR$ occurs in multiple steps involving many intermediates, among which *CO is an important chemical species. Notably, the *CO desorption results in the CO formation; protonated *CO evolves in $CH_4$, whereas the dimerization of two *CO intermediates leads to the formation of $C_2$ products[15,17,29]. Therefore, to unveil the intrinsic adsorption behavior of the *CO intermediates, diffuse reflectance infrared Fourier-transform spectroscopy (DRIFTS) was conducted on the heteronuclear/homonuclear diatomic COFs under 53 °C. In this method, the COFs were first subjected to CO saturation adsorption and then desorption under a flowing $N_2$ atmosphere to remove the CO adsorbed on the catalyst sites. As expected (Fig. 4d, e and Supplementary Fig. 34), the strongest desorption peaks are observed at 2070 and 2117 $cm^{-1}$ at the three catalysts for the absorbed CO[5,9]. Importantly, the stretching vibration band (-2165–2230 $cm^{-1}$) can be ascribed to the *OC–CO intermediate produced by two CO dimerization[11,29], which is much more intensive in ZnPor-RuCuDAC than in ZnPor-Ru$_2$DAC and ZnPor-Cu$_2$DAC as further evident from the

2D contour plots of these three COFs (Fig. 4f), indicating that, unlike the homonuclear sites, the heteronuclear Ru–Cu sites exhibit synergistic effects on the *CO dimerization[5,17,28]. With $N_2$ purging, the peak intensity gradually decreases due to the removal of the adsorbed CO. Noteworthily, compared with ZnPor-Cu$_2$DAC (Supplementary Fig. 34), two residual CO adsorption peaks are always detected in the DRIFTS of ZnPor-RuCuDAC and ZnPor-Ru$_2$DAC, irrespective of their $N_2$ cleaning time, suggesting that the Ru-containing COF skeleton exhibits more robust *CO chemical interaction[15], particularly in ZnPor-Ru$_2$DAC. Furthermore, the band center energies, $\varepsilon_{d,Ru}$ and $\varepsilon_{d,Cu}$ of the density of states (DOS) were also surveyed to evaluate the CO adsorption ability. Evidently (Fig. 4g and Supplementary Fig. 35), $\varepsilon_{d,Ru}$ (−0.73) of ZnPor-RuCuDAC shifts toward the Fermi level ($E_f$), more prominent than ZnPor-Ru$_2$DAC (−1.77). Also, $\varepsilon_{d,Cu}$ (−0.92) of ZnPor-RuCuDAC moves toward the $E_f$ as compared to that of ZnPor-Cu$_2$DAC (−2.25). A closer $\varepsilon_{d,Ru}$ to $E_f$ confirms the lower adsorption energy for the absorbate on the Ru–Cu sites than on the homonuclear sites[30]. This indicates the existence of a synergistic heteronuclear diatomic effect, which enhances the adsorption and C–C coupling of CO molecules over the ZnPor-RuCuDAC.

To reveal the $CO_2RR$ intermediates and further investigate the diatomic-type role on the photoreduction, $CO_2RR$ in situ DRIFTS experiments were performed at different irradiation times. As evident

from Fig. 4h, the peak at 1342 cm$^{-1}$ can be assigned[11] to the stretching of absorbed b-CO$_3^-$, and the symmetric stretching and asymmetric stretching vibration peaks of HCO$_3^-$ are observed at 1445 and 1676 cm$^{-1}$, respectively[8]. The increase in the intensity of the infrared signal at 1587 cm$^{-1}$ with the increasing irradiation time can be attributed to the ˙COOH group[5,8], which is a pivotal intermediate for CO$_2$ conversion to CO and other solar fuels. The peaks at 1372 and 1496 cm$^{-1}$ can be assigned to ˙OC–CO[28], which is a typical ˙CO coupling product and a crucial intermediate for acetate synthesis. The peaks at 1251 and 1396 cm$^{-1}$ belong to the $\delta$(C–H) and $\delta$(C–C) deformation vibrations of the hydrogenated intermediates after the C–C coupling, whereas the peak at 1552 cm$^{-1}$ can be ascribed to the ˙COO stretching vibration of CH$_3$COOH[17,29]. Compared with ZnPor-RuCuDAC, the in situ DRIFTS profiles of both ZnPor-Ru$_2$DAC and ZnPor-Cu$_2$DAC indicate similar reaction intermediates (Supplementary Figs. 36, 37). However, as further seen in Fig. 4i, the intermediate intensity of the formed acetate is significantly stronger during the photoreduction on ZnPor-RuCuDAC than on ZnPor-Ru$_2$DAC and ZnPor-Cu$_2$DAC. Moreover, no obvious ˙CO infrared peaks are observed in the spectrum of ZnPor-RuCuDAC, but ZnPor-Cu$_2$DAC exhibits a stronger ˙CO infrared desorption than ZnPor-Ru$_2$DAC at 1884 cm$^{-1}$ during the photocatalytic process. This result indicates that a large amount of CO is produced on the surface of ZnPor-Cu$_2$DAC than on those of the other two COFs. These differences support that the ˙CO intermediates on Ru–Cu sites are prone to couple and form ˙OC–CO intermediates and then acetate, whereas the ˙CO intermediates on the homonuclear Ru–Ru and Cu–Cu sites are desorbed to form CO molecules easily.

Based on the described experimental evidences, there are two possible reaction pathways for acetate formation after the C–C coupling, as shown in Supplementary Fig. 38. For the first pathway, the two oxygen atoms of the evolved acetate are derived from CO$_2$, whereas for the second, one of the oxygen atoms originates from CO$_2$ and the other from H$_2$O. To identify the actual route, an isotopic labeling experiment was conducted with 10 vol. % H$_2^{18}$O (Supplementary Fig. 39). The corresponding mass spectrum shows that no $^{18}$O-labeled acetate product (CH$_3$C$^{16}$O$^{18}$OH or CH$_3$C$^{18}$O$^{16}$OH) is detected, indicating that the conversion of CO$_2$ to acetate occurs via the first pathway (Supplementary Fig. 38a). To confirm the mechanism of CO$_2$RR to acetate, Fig. 5a illustrates the Gibbs free energy diagrams of the deduced reaction pathways over these diatomic COFs. As described from step I to step IV, two ˙CO formation pathways on the initial active sites of Ru–Cu ($\Delta G = 0.61$ eV), Ru–Ru ($\Delta G = 0.83$ eV), and Cu–Cu ($\Delta G = 1.35$ eV), appear to be endothermic in nature. Notably, the first CO$_2$ activation to form the ˙COOH and ˙CO intermediates is much easier than the second one, which produces ˙CO + ˙COOH and ˙CO + ˙CO on the diatomic sites. Intriguingly, the coupling of the (˙CO + ˙CO) intermediate to form ˙OC–CO does not involve electron or proton transfer and is essentially a thermal process. However, the C–C coupling energy barrier is much lower for the Ru–Cu (0.51 eV) site than for Ru–Ru (1.55 eV) and Cu–Cu (2.18 eV). According to $\Delta G$ comparison of different reaction pathways for adsorbed ˙CO + ˙CO (Supplementary Figs. 40–42), the ˙OC–CO intermediates, formed on heteronuclear Ru–Cu sites, are more thermodynamically favored than the ˙CO desorption and ˙CO hydrogenation to *CHO or *COH. In contrast, the $\Delta G$ of ˙CO desorption on ZnPor-Ru$_2$DAC is smaller than that for ˙CO protonation (Supplementary Fig. 41), and ZnPor-Cu$_2$DAC promotes an exothermic CO formation rather than an endothermic C–C coupling (Supplementary Fig. 42). Moreover, the immediate hydrogenation of ˙OC–CO to ˙OC–COH on these diatomic site COFs is exothermic, and consequently, the elementary steps involved the reduction of ˙OC–COH to ˙HO(COC)CH$_2$ are all downhill, followed by the endothermic steps of ˙HO(COC)CH$_2$ → ˙HOOC–CH$_3$ → HOOC–CH$_3$ in the free energy change diagrams. From CO$_2$ to CH$_3$COOH, the overall catalytic system is exothermic (−0.31 eV) on heteronuclear Ru–Cu site, while the catalytic system of ZnPor-Ru$_2$DAC (0.23 eV) and ZnPor-RuCuDAC

(0.69 eV) is endothermic. Therefore, the ˙CO intermediate on ZnPor-Cu$_2$DAC is inclined to desorb as CO rather than to couple into ˙OC–CO as it does on the Ru–Cu site. This inferior C–C coupling accounts for the very low yield of acetate on homonuclear diatomic COFs.

## Charge densities and orbital interactions

To investigate the diatomic-type effect on the electron density distribution in the ground- and excited-state active sites, we calculated the charge distribution for the heteronuclear and homonuclear sites, respectively. As shown in Supplementary Fig. 43, the ground-state electrons of these three COFs are mostly delocalized over the ZnPor cores, but excited-state charge density distributions vastly enrich on the diatomic sites, indicating the occurrence intraskeletal photo-generated energy transfer and electron diffusion. Moreover, as depicted in Supplementary Fig. 44, the Bader charges of the excited-state Ru atoms are 10.32e (ZnPor-RuCuDAC) and 8.75e (ZnPor-Ru$_2$DAC), but the Cu atom Bader charges are 5.61e (ZnPor-RuCuDAC) and 6.28e (ZnPor-Cu$_2$DAC). An increase in the Bader charge of the Ru-containing sites reveals the preferable accumulation of photoelectrons, which promote strong interactions between the reactive sites and the adsorbed CO$_2$ molecules as well as lead to the formation of reaction intermediates. In addition, the Ru–Ru and Cu–Cu sites show equilibrium Bader charge distributions, whereas ZnPor-RuCuDAC exhibits a difference of 4.71 between the Bader charges of the Ru and Cu atoms in the Ru–Cu site. This indicates that the heteronuclear diatomic sites can increase the asymmetry of the electron delocalization, resulting in longer Ru–Cu bonds (2.71 Å) than Ru–Ru (2.67 Å) and Cu–Cu (2.61 Å) bonds.

To analyze the low C–C coupling reaction barrier of the Ru–Cu sites, we calculated the charge density difference on these diatomic COFs after the formation of two ˙CO intermediates. The two ˙CO intermediates nearly appear parallel to each other on the heteronuclear Ru–Cu atoms with a side-to-side configuration and a negligible 4.7° twist along the dual-atom direction (Supplementary Fig. 45). This twist is different from the broad dihedral angle distortion of the Ru–Ru (19.5°) and Cu–Cu (27.1°) sites (Supplementary Figs. 46, 47), indicating a weak molecular repulsion between the two ˙CO intermediates on the Ru–Cu site. Concurrently, the metal–metal bonds are shortened to 0.06 Å (Ru–Cu), 0.01 Å (Ru–Ru), and 0.02 Å (Cu–Cu) compared to the excited-state bond lengths. Due to the excited-state asymmetric charge distribution of ZnPor-RuCuDAC, the Bader charges difference value of carbon-to-carbon and oxygen-to-oxygen atoms in the two ˙CO intermediates formed at the Ru–Cu sites are 2.95e (for C) and 2.76e (for O), which are different from the equal Bader charge of the homonuclear Ru–Ru and Cu–Cu centers. Thus, to achieve a stable chemical microenvironment, the different carbon-to-carbon and oxygen-to-oxygen Bader charges distributed on the neighboring ˙CO intermediate promote C–C coupling and suppress the electrostatic repulsion between the two ˙CO intermediates, resulting in weak dipole–dipole interactions. As a result, the reaction energy barrier for ˙CO + ˙CO → ˙OC–CO is relatively low at the Ru–Cu sites. Additionally, the differential charge density maps show that compared to the homonuclear COFs, heteronuclear Ru–Cu exhibit a significant fraction of crossed electron cloud (Fig. 5b–d and Supplementary Fig. 48), and the carbon-to-carbon distance (1.59 Å) of the two ˙CO intermediates formed at the Ru–Cu sites is shorter than that of Ru–Ru and Cu–Cu (Supplementary Figs. 45–47). These findings can collectively augment the collision probability of the two ˙CO intermediates and promote the coupling of the adjacent ˙CO intermediates to form ˙OC–CO. After the C–C coupling (Supplementary Fig. 49), the Bader charges of the carbon-to-carbon and oxygen-to-oxygen atoms in the ˙OC–CO intermediate formed at the Ru–Cu site are reduced to 0.08e (for C) and 0.18e (for O). In addition, the Ru–Cu bond is further shortened to 0.07 Å compared to those of Ru–Ru (0.03 Å) and Cu–Cu (0.02 Å). This bond length reduction can be

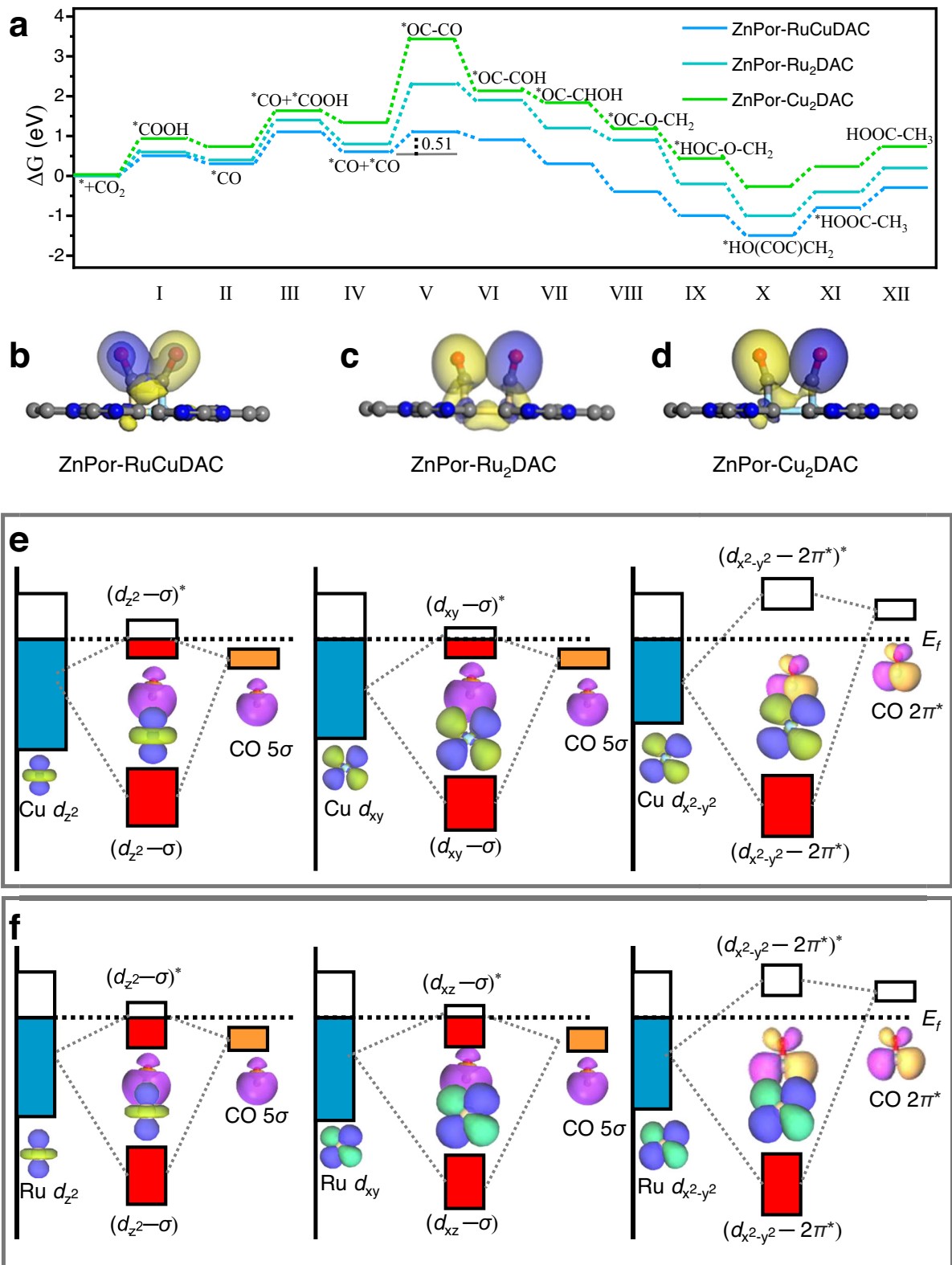

**Fig. 5 | Theoretical analyses of acetate formation during CO₂RR of COFs. a** Gibbs free energy diagrams of CO₂ photoreduction to acetate pathway of ZnPor-RuCu-DAC, ZnPorRu₂DAC, and ZnPorCu₂DAC. **b**–**d** Differential charge density maps of **b** ZnPor-RuCuDAC, **c** ZnPorRu₂DAC, and **d** ZnPorCu₂DAC. **e**, **f** Schematic illustration of the adsorbed CO ($5\sigma$, $2\pi^*$) orbital interactions with the **e** Cu $3d$ and **f** Ru $4d$ orbitals in ZnPor-RuCuDAC.

ascribed to the intensive coupling interactions of *CO on the Ru−Cu heteroatom.

Considering that the Bader charge arrangement and spatial structure of *CO for these diatomic COFs are dependent to the

interactions between the metal sites and *CO, orbital coupling behaviors of the metal active center and *CO intermediate were further scrutinized. Supplementary Figs. 50–53 show the calculated projected DOS (PDOS) of the adsorbed CO ($5\sigma$ and $2\pi^*$) and metal-$d$ orbitals ($d_{z^2}$,

$d_{xz}$, $d_{xy}$, $d_{yz}$, and $d_{x^2-y^2}$) in these COFs. Compared to the homonuclear COFs, the Ru-4$d$ and Cu-3$d$ states of the heteronuclear ZnPor-RuCuDAC are almost delocalized, especially the Ru-4$d_{xy}$ and Cu-3$d_{z^2}/d_{xy}$ orbitals. Such delocalization of the Ru and Cu $d$ states can be understood by analyzing the excited-state orbital interactions between the Ru and Cu atoms. In the ZnPor-RuCuDAC, the Ru-4$d_{z^2}$ and Cu-3$d_{z^2}$ orbitals resonate at −0.5 and −7.85 eV, and Ru-4$d_{x^2-y^2}$ and Cu-3$d_{x^2-y^2}$ resonate at 1.46 eV, indicating a strong 4$d$–3$d$ gradient orbital coupling at the heteroatoms. The strong interaction between Ru and Cu atoms in the hetero-diatomic catalyst can lower the orbital energy levels and reduce the electron delocalization, resulting in a changeable *CO absorption behavior. In ZnPor-RuCuDAC, the Ru-4$d$ PDOS mean-ingfully overlap with the *CO molecular orbital by (Ru, $d_{z^2}/d_{xz}$) − (CO 5$\sigma$) and (Ru, $d_{xy}/d_{yz}/d_{x^2-y^2}$) − (CO 2$\pi$*) interactions (Supplementary Fig. 50), and the Cu-3$d$ site exhibits an analogous orbital coupling via (Cu, $d_{z^2}/d_{xy}$) − (CO 5$\sigma$) and (Cu, $d_{xz}/d_{yz}/d_{x^2-y^2}$) − (CO 2$\pi$*) interaction (Supplementary Fig. 51). However, no perceivable overlapping is recognized in both (Ru, 4$d$) − (CO, 5$\sigma$/2$\pi$*) of ZnPor-Ru₂DAC and (Cu-3$d$) − (CO, 5$\sigma$/2$\pi$*) of ZnPor-Cu₂DAC (Supplementary Figs. 52, 53). The strongly overlapped orbitals split into bonding and antibonding orbitals, giving rise to decreasing energy splitting levels of the bonding and antibonding states of (Ru/Cu, $d$) − (CO, 5$\sigma$/2$\pi$*) in ZnPor-RuCuDAC, as compared to ZnPor-Ru₂DAC and ZnPor-Cu₂DAC (Fig. 5e, f and Supplementary Fig. 54). As a result, the Ru-*CO and Cu-*CO sites possess more stable chemical structures because of large electron occupancies of the corresponding bonding and antibonding orbitals. Thus, the two *CO intermediates exhibit a small dihedral angle distortion and a short carbon-to-carbon distance at the Ru–Cu sites.

The *OC–CO protonation is considered another crucial rate-limiting step during CO₂RR, and therefore, crystal orbital overlap population (COOP) calculation was conducted to elucidate the underlying mechanism of the reduced protonation energy barrier in the diatomic sites (Supplementary Fig. 55). The COOP between the C and O atoms of *CO (Supplementary Fig. 56a, b), in the *OC–CO intermediate adsorbed on the Ru (Ru-*CO) or Cu (Cu-*CO) site of ZnPor-RuCuDAC, displays typical bonding and antibonding orbital populations; the fraction of antibonding population is more around the $E_f$. The intensity of the *CO antibonding population across the $E_f$ is much stronger on the Cu-*CO site than the Ru-*CO site in ZnPor-RuCuDAC. The *CO antibonding population of *OC − CO in the $E_f$ at ZnPor-Cu₂DAC also reinforces as referred to ZnPor-Ru₂DAC (Supplementary Fig. 56c, d). These results reveal that for the *CO absorbed on Cu site, the C=O bond of OC−CO* is much weaker and can be easily activated, which is favorable for protonation and thus reduce the potential barrier for *OC−COH formation. The reciprocal action between the adsorbed *OC−COH and the metal (Ru, Cu) sites was also evaluated (Supplementary Fig. 57) based on the COOP between C of OC−COH* and the metal atoms in these COFs. In ZnPor-RuCuDAC, the antibonding population of the Cu-C bond in Cu-*COH increases in the valence bands (below $E_f$); no such effect is observed for that of the Ru-C bond in Ru-*CO (Supplementary Fig. 58a, b). The same behavior is also observed in ZnPor-Ru₂DAC and ZnPor-Cu₂DAC (Supplementary Fig. 58c–f), indicating the weaker bond strength between the metal atom and C of COH*. Subsequently, the integrated COOP (ICOOP) was calculated by merging the energy integral with the highest occupied band[48]. The ICOOP values of the Ru−C (Ru-*CO) and Cu−C (Cu-*COH) bonds are 0.32 and 0.17 in ZnPor-RuCuDAC, respectively. Similarly, both ZnPor-Ru₂DAC and ZnPor-Cu₂DAC also exhibit higher ICOOP values of the metal atom and C of *CO than that of the metal atoms and C of *COH. The more positive ICOOP value of the metal−C bond implies a more stable adsorption configuration and stronger interactions between the metal and *CO. By contrast, the lower ICOOP values indicate that the metal−C bond can be easily cleaved.

The results of photocatalytic performance, in situ spectra analyses, and density functional theory calculations of these diatomic sites collectivity reveal the superior acetate selectivity of ZnPor-RuCuDAC. First, the Ru-Cu heteroatom shows a strong *CO adsorption. Second, the Ru-4$d$–Cu-3$d$ orbital resonance causes a strong gradient orbital coupling, which results in a much weaker repulsive force between the two absorbed *CO intermediates on Ru−Cu het-eroatom; consequently, an asymmetric charge distribution is observed, and a side-to-side absorption behavior and narrow dihedral angle distortion become evident. Third, the strong overlapping between the Ru/Cu-$d$ and CO molecular orbitals (5$\sigma$ and 2$\pi$*) results in decreasing energy splitting levels of the bonding and antibonding states of (Ru/Cu, $d$) − (CO, 5$\sigma$/2$\pi$*) in ZnPor-RuCuDAC. These results can enhance the collision probability of *CO intermediates and pro-mote the coupling of the adjacent *CO intermediates to generate acetate on Ru−Cu heteroatoms.

In summary, novel ZnPor-RuCuDAC matrices were engineered as new PMD-like heterogeneous catalysts to bridge single-atom photosensitizers and diatomic catalytic cores. The heteroatomic ZnPor-RuCuDAC exhibited the highest acetate selectivity of 95.1 %, whereas that of ZnPor-Cu₂DAC and ZnPor-Ru₂DAC were found to be 36.9 and 21.6%, respectively. Further, ZnPor-Cu₂DAC exhibited the most superior CO selectivity. In situ characterizations and theoretical calculations reveal that the distinct charge distributions of hetero-nuclear Ru−Cu sites suppress the electrostatic repulsion between the two neighboring *CO intermediates, resulting in a weak dipole−dipole interaction and promoting the C−C coupling process. The reaction *CO + *CO → *OC−CO progressed more easily at the heteroatomic Ru−Cu sites than at Ru−Ru and Cu−Cu. Therefore, the heterogenization of COF matrices, with functional and structurally superior diatomic sites, offers significant insights toward symmetry-forbidden coupling mechanism, which can aid in regulating the C₁ intermediate coupling strength by manipulating the dual-atom metal gradient orbital interactions for various photo(electro)catalysis applications such as CO₂ reductions, hydrogen productions, oxygen evolutions, and nitrogen fixation.

## Methods

### Materials
All analytical-grade raw materials (benzene-1,3,5-tricarboxaldehyde, neopentyl glycol, $p$-toluenesulfonic acid monohydrate, Pyridine-2,6-dicarboxylic acid, 1,2-phenylene diamine) were purchased from Sigma-Aldrich Chemicals and directly used without further purification. Dehydrated solvents were obtained after treating solvents with stan-dard procedures.

### Synthesis of 5,10,15,20-Tetrakis(3,5-dibenzaldehyde) por-phyrin Zn(II)
5,10,15,20-Tetrakis(3,5-dibenzaldehyde) porphyrin Zn(II) (ZnPor) was obtained by heating 5,10,15,20-Tetrakis(3,5-dibenzaldehyde) porphyrin[32,33] (0.50 mmol) and zinc acetate dihydrate (1.00 mmol) in $N$,$N$-dimethylformamide (DMF) solution at 165 °C for 10 h under N₂ atmosphere. After filtration of the solvent, the product was washed with deionized water and further purified by column chromatography to obtain the purple ZnPor powder (Yield: 418.5 mg, 92%. TOF-MS (m/z) calcd. for C₅₂H₂₈O₈N₄Zn [M + H]⁺ 900.12, found 899.83).

### Synthesis of 2,6-bis(5-amino-1H-benzimidazol-2-yl)pyridine Ruthenium(II) or Copper (II) (RuN₃ or CuN₃) pincer complex
[RuCl₂(p-cymene)]₂ was prepared via refluxing RuCl₃ (1.00 mmol) and dry ethanol (5 mL) under N₂ atmosphere for 12 h, which was purified by filtration and washed with CHCl₃ and hexane. Then, 2,6-bis(5-amino-1H-benzimidazol-2-yl)pyridine[34,35] (0.36 mmol) and [RuCl₂(p-cymene)]₂ (0.72 mmol) were dissolved in DMF that was stirred at 100 °C for 24 h under N₂ atmosphere. After removing the solvent by reduced pressure distillation, the crude product was purified by die-thyl ether and tetrahydrofuran, followed by overnight vacuum drying

(Yield: 101.2 mg, 90.6%. TOF-MS (m/z) calcd. for $C_{19}H_{15}ClN_7Ru$ [M + H]$^+$ 478.11, found 477.65). Similarly, $CuN_3$ pincer complexes are prepared with the same procedure by using [$CuCl_2$(p-cymene)]$_2$ precursors.

## Synthesis of ZnPor-RuCuDAC COF

The hydrazone-linked ZnPor-RuCuDAC COF was prepared via condensation of ZnPor, $RuN_3$, and $CuN_3$ monomers (Supplementary Fig. 1). Briefly, a Pyrex tube was added with ZnPor (0.05 mmol), $RuN_3$ (0.10 mmol), $CuN_3$ (0.10 mmol), o-dichlorobenzene (4.5 mL), n-butanol (4.5 mL), and acetic acid solution (1.0 mL, 5.0 M). After degassing by three freeze-pump-thaw cycles and sealed under vacuum, the tube was heated at 150 °C for 7 days in the oven. The resulting precipitate was collected by centrifugation and exhaustively washed by Soxhlet extractions with methanol and tetrahydrofuran for 72 h. Finally, the brown ZnPor-RuCuDAC COF was dried under a vacuum oven at 80 °C, with a yield of 82.7% on the basis of the ZnPor monomer. For comparison, the counterparts with different Ru:Cu molar ratio (Ru:Cu = 0:10, 1:9, 3:7, 5:5, 7:3, 9:1, and 1:0) were prepared via varying the addition amounts of $RuN_3$ and $CuN_3$ monomer.

## Photocatalytic $CO_2$ reduction assessments

A typical $CO_2$ photoreaction experiment was performed by dispersing catalysts (8 mg) into a mixture solution (12 mL acetonitrile, 2 mL deionized water, and 1 mL triethanolamine) at homemade quartz reactor (230 mL). The reactor was purged with purity $CO_2$ (99.999%) after three cycles of vacuum and refilling. Then, the photosystem was sealed and illuminated with a 150 W Xe-lamp (Abet Technologies) irradiation. Especially, the intensity of Xe-lamp light was calibrated to be 100 mW cm$^{-2}$ to the quartz reactor surface (calibrated by an CEL-NP2000 Optical Power Meter), and the irradiation area was 5.65 cm$^2$. With Hamilton syring assistance, the 0.5 ml gas products were quantified by gas chromatography (8890, Agilent Technologies) equipped with a TCD detector (5 A molecular sieve column) and a FID detector (TDX-01 column) connected to the methane reforming furnace using Ar as the carrier gas. The liquid product was analyzed by high-performance liquid chromatography equipped with a capillary column. In the photoreaction system, hydrogen ($H_2$), carbon monoxide (CO), formic acid (HCOOH), and acetate ($CH_3COOH$) are the main products, and no other products such as $CH_4$, $CH_3OH$, $C_2H_4$, and $C_2H_6$ is found. For the cyclic test, the used catalyst was recovered by centrifugation after the first reaction, which was then applied in the second reaction test. At the second reaction end, the catalyst was retrieved again for the third test, and so on. This process was repeated four times.

The selectivity for $CO_2$ reduction to acetate, formate, CO, and $H_2$ (8e$^-$ for the formation of acetate, 2e$^-$ for formate, 2e$^-$ for CO, and 2e$^-$ for $H_2$) were calculated according to the following equations:

$$Sensitivity_{acetate}(\%) = \frac{8n_{acetate}}{\left(8n_{acetate} + 2n_{formate} + 2n_{CO} + 2n_{H_2}\right)} \times 100 \tag{1}$$

$$Sensitivity_{formate}(\%) = \frac{2n_{formate}}{\left(8n_{acetate} + 2n_{formate} + 2n_{CO} + 2n_{H_2}\right)} \times 100 \tag{2}$$

$$Sensitivity_{CO}(\%) = \frac{2n_{CO}}{\left(8n_{acetate} + 2n_{formate} + 2n_{CO} + 2n_{H_2}\right)} \times 100 \tag{3}$$

$$Sensitivity_{H_2}(\%) = \frac{2n_{H_2}}{\left(8n_{acetate} + 2n_{formate} + 2n_{CO} + 2n_{H_2}\right)} \times 100, \tag{4}$$

where, $n_{acetate}$, $n_{formate}$, $n_{CO}$, and $n_{H_2}$ are the mole amounts of produced acetate, formate, CO, and $H_2$, respectively.

## Data availability

The data presented in this article are available from the corresponding authors on reasonable requests.

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

## Acknowledgements

This work was supported by the National Research Foundation of Korea (NRF) grants funded by the Ministry of Science and ICT (2022R1A2C3003081). This work was also supported by Samsung Science & Technology Foundation fund by Samsung Electronics (SSTF-BA1702-07).

## Author contributions

J.M.W. and T.K.K. conceived the project. J.H.L. and W.-D.J. synthesized the samples. J.M.W. and Q.Y.Z. performed basic optical characterizations and photocatalytic assessments. T.G.W. performed STEM and EDS mapping characterizations. Y.X.Z. performed theoretical calculations. J.M.W. and T.K.K. led the analyses of the data with contributions from all authors. J.M.W. and T.K.K. wrote the paper with input from all authors.

## Competing interests

The authors declare no competing interests.
