## [Peer Review File · Nature Communications]

Asymmetric Gradient Orbital Interaction of Hetero-Diatomic Active Sites for Promoting C–C CouplingREVIEWER COMMENTS

Reviewer #1 (Remarks to the Author):

The authors reported the preparation of Zn-porphyrin/RuCu-pincer complex DAC (ZnPor-RuCuDAC), which were employed as a photocatalyst for CO₂RR. They tried to make a complete story from the synthesis to the reaction study and theoretical elucidation of the mechanism. The presented results and strategy of the photocatalytic system is interesting, however there are few comments to be clarified before it considers for publication:

The authors thought the formation of Cu–Ru, Cu–Cu in the ZnPor-RuCuDAC matrices based on the EXAFS results. The authors should clarify it by multiple characterization methods.

In the EXAFS data with fitting, could the authors observe the Cu-N, Ru-N and so on bonds using any characterization methods? What was the actual structure of the Cu Ru, Zn with N? Besides, it was very strange that the XPS peaks were very sharp, although the EXAFS were ambiguous to analyze.

For the photocatalytic CO₂ reduction process, the transfer and life of the excited electrons can be the time-resolved PL and fs-TA absorption spectra, which can obtain the migration mechanism of photogenerated charge-holes in ZnPor-RuCuDAC matrices during the CO₂ reduction process.

The characterization of ZnPor-RuCuDAC is not enough to understand the material's state. The authors reported XANESs, and atomic-resolution STEM with only a few snapshots without citations. What were the chemical properties of ZnPor-RuCuDAC? Can the authors prepare more direct evidence for proving the Ru–Cu sites?

The authors need to provide detailed proposed reaction mechanism figure with proper band edge and band gap values to understand the charge transfer process and correlate with existed results.

Reviewer #2 (Remarks to the Author):

This work synthesized the diatomic-site COF catalyst, ZnPor-RuCuDAC, by bridging Zn-porphyrin and Ru/Cu-pincer complexes. Compared with the homoatomic counterparts ZnPor-Cu₂DAC and ZnPor-Ru₂DAC, it shows a best acetate selectivity in photocatalytic CO₂ reduction reaction. The author believes that this is due to the distinct charge distributions of heteronuclear Ru-Cu sites, which suppress the electrostatic repulsion between the two neighboring *CO intermediates, resulting in a weak dipole-dipole interaction and promoting the C-C coupling process. The production of acetate was deeply analyzed by many in-situ experiments and calculations. However, there are some problems in the analysis of the ZnPor-RuCuDAC structure, including the framework structure of COF and the local structure of active sites. This may affect the establishment of the model in the subsequent calculation,

resulting in the deviation of the calculation results.

1. How can the authors get the specific ZnPor-RuCuDAC COF structure in Supplementary Scheme 1? If it is only a conjecture or schematic diagram, whether the description in the element analysis results “the experimental weight percentage of elemental C, H, N, Zn, Ru and Cu are closed to their respective theoretical compositions” is not logical, because the author has not clearly given the skeleton structure here.
2. The name of the corresponding catalysts should be clearly marked in Figure 1b-d.
3. The spatial atomic cross-distribution of Zn, Ru and Cu elements on the ZnPor-RuCuDAC substrate in Figures 1g-j cannot match the AB-staggered stacking model in Supplementary Figure 4. In the AB-staggered stacking model in Supplementary Figure 4, the Zn element and Ru/Cu elements cannot appear at the same position when viewed from a vertical perspective. The inset in Figure 1g also shows the same information. This means that there may be problems with the skeleton structure of COF given in the article.
4. Page 6, “The elemental analysis results of ZnPor-RuCuDAC single crystal (Supplementary Fig. 8c-f) show a uniform distribution of the Zn, Ru, and Cu elements on the scaffolds.” Note that this material is not a single crystal.
5. Does “the rising edges” means edge energy? It is suggested that the authors mark the edge positions as “edge energy” or “pre-edge energy” in Supplementary Table 6 for easy identification. In addition, the representation of Figures 2a-c cannot effectively identify the difference between different catalysts. Is the author considering a different form of expression?
6. The Zn-Zn peak in Figure 2d is labeled incorrectly.
7. The wavelet transform provides contributions of near neighbors in both R-space and k-space, and its abscissa is related to the atomic weight. Atoms with small atomic numbers have weak scattering ability to photoelectrons, and their strongest oscillation will occur in the lower k part, while the opposite is true for atoms with large atomic numbers. However, the comparison between Figure 2g and Figure 2i shows that Ru with a larger atomic number corresponds to a smaller k value. This is not reasonable. The same problems apply to the analysis of ZnPor-RuCuDAC and ZnPor-Cu2DAC, as well as a large number of the wavelet transform analyses in Supplementary Information. In addition, Ru-N in ZnPor-RuCuDAC and ZnPor-Ru2DAC or Cu-N in ZnPor-RuCuDAC and ZnPor-Cu2DAC should have extremely approximate values. This part of EXAFS data needs more accurate analysis.
8. The mark of Supplementary Figure 10a is incorrect.
9. According to the structure (Supplementary Figure 4) and the unit cell parameters given by the author, the distance between the Ru and Cu/Ru atoms in the upper and lower layers also seems to be very close (3.81 or 3.81/2), and all Ru and Cu atoms are in the state of overlapping. Why is this signal not observed in EXAFS? Does this also mean that the author's analysis of the COF skeleton structure needs further verification?
10. How does the author determine that only Ru-Cu pairing exists in ZnPor-RuCuDAC, without Ru-Ru pairing and Cu-Cu pairing? The EXAFS fitting results show that the length of the Ru-Cu bond in ZnPor-RuCuDAC is 2.70, and the length of the Ru-Ru bond in ZnPor-Ru2DAC is 2.68. This seems indistinguishable by fitting.
11. In the calculation of band structures, the values (1.05, 0.95, and 0.93 eV) obtained from the SRPES valence band spectra (Supplementary Figures 13a-c) should be the energy difference between the VBM

and Fermi level (EVBM-Ef), not the EVBM. The band structure analysis needs to be corrected.

12. This new band may be attributed to the deformation vibration of Ru-Cu sites, containing the absorbed C-C coupling intermediates during CO₂RR. Whether there is a certain proof basis for the description of the C-C coupling intermediates, it is suggested to consider citing literature.

13. It is recommended to add literature support for signal attribution in DRIFTS.

14. When the interaction between Ru-Cu sites during the photocatalytic process is calculated, why not consider the interaction of active sites between adjacent layers?

15. It seems that the author synthesized a COF material after reading the full text. Why is it called an intraskeletal photogate molecular device (PMD) catalyst?

Reviewer #3 (Remarks to the Author):

This manuscript reported photocatalytic CO₂ reduction on Ru-Cu couple site on COF and the results are highly interesting because acetic acid was selectively formed in spite of difficulty of C-C bond formation by photocatalyt. The content is well organized however the following points are still not clear and revision is required on the following points.

1) Stability is still wondering because COF is consisted of Zn-porphyrin and pyridine with dehydration reaction. Since photocatalytic reaction is performed under coexistence of triethanolamines which may be alkaline condition, hydration of this C=N bond may expect to be dissociated during reaction. Although authors show XRD after reaction, broadening of the peaks was recognized and also please add intensity for Y axis in supplementary Fig.20 (b).

2) Cl impurity may be contained, and I wonder Cl content in catalyst may be negative impact or prevent further oxidation of acetic acid.

3) P.12, Authors claim Ru-Cu stretching and bending mode were observed at 226 and 252 cm⁻¹, respectively in Raman spectroscopy without any proof and so I suggest authors to simulate Raman spectrum based on the model proposed in Supplementary Scheme 1 for demonstrating. Raman spectroscopic data after measurement is also required to confirm Ru-Cu DAC sustained.

4) Fig.4, Temperature is missing and if this is room temperature, adsorption of CO is quite weak and so I wonder such weak adsorption species can contribute to reaction.

5) Fig.5 and discussion on reaction route; It seems CO coupling is the rate determining step and CO hydrogenation is not large activation energy required. If so, why CH₄ or coupling of CH₃ species of C₂H₆ are not formed. From energy level calculation, CH₃COOH energy level is almost the same with that of CO₂ and this is highly strange and not reliable.

Also, adsorption model of CO on Ru and Cu DAC site is same direction and what data does support of this adsorption model and why such orientation is occurred because opposite site seems to be more reasonable form energy minimization.

6) No detail information of oxidation species and I think triethanolamine may be used for sacrificial agent, however, if so, ethanol may form and further oxidized to CH₃COOH. However, from tracer experiment using ¹³C, the authors claimed only ¹³C contained CH₃COOH was formed and if so, what is oxidation species and how to consume hole?

- 7) In experiment, the authors used acetonitrile and why acetonitrile is used for solvent? If simple water is used for solvent, what product is obtained?
- 8) Change of oxidation number of Ru and Cu seem to be important to determine the activity and so please measure the XPS for analysis of oxidation state of Ru and Cu before and after reaction.
- 9) Supplementary Figure 6, pore size distribution is shown, and I think wall thickness of COF can be estimated from the observed pore size and this is required for analysis of pore structure of the obtained COF. What pore is corresponding to 5 nm which is weak peak in pore size distribution measurement?
- 10) Supplementary Figure 12, The authors estimated band gap from UV-VIS., but it seems estimated band gap is overestimated and should estimate from straight line in more wide area. In this case, the estimated narrow band gap should be discussed.

Responses to Reviewers (NCOMMS-23-00556A)

Reviewer #1 (Remarks to the Author): *The authors reported the preparation of Zn-porphyrin/RuCu-pincer complex DAC (ZnPor-RuCuDAC), which were employed as a photocatalyst for CO₂RR. They tried to make a complete story from the synthesis to the reaction study and theoretical elucidation of the mechanism. The presented results and strategy of the photocatalytic system is interesting, however there are few comments to be clarified before it considers for publication:*

1. The authors thought the formation of Cu–Ru, Cu–Cu in the ZnPor-RuCuDAC matrices based on the EXAFS results. The authors should clarify it by multiple characterization methods.

Re: Thanks a lot for this professional comment.

To further confirm the Ru and/or Cu diatomic pairs within these COF matrices, we have conducted the high-resolution X-ray photoelectron spectra (XPS) characterization of Zn, Ru and Cu elements on the revision for chemical microenvironment observation. All samples are followed an Ar etching process before XPS analysis.

As seen in the revised supplementary information, the survey XPS (Supplementary Fig. 13a) indicate that Zn, Ru, Cu, C and N coexist in the precursors and/or these COFs. The high-resolution Zn2p XPS spectra (Supplementary Fig. 13b) show that ZnPor have Zn(II) 2p_{3/2}/2p_{1/2} binding energy (BE) peaks of 1021.81/1044.95 for Zn-N bond, which show a positive shift for ZnPor-RuCuDAC (~0.18 eV), ZnPor-Ru₂DAC (~0.22 eV) and ZnPor-Cu₂DAC (~0.23 eV), ascribable to a decreased electron density of the ZnPor cores.

The high-resolution Ru3p XPS spectrum (Supplementary Fig. 13c) of RuN₃ exhibits two BE peaks at 462.57 (Ru3p_{3/2}) and 485.83 (Ru3p_{1/2}) eV for Ru-N bond, belonging to the Ru(II) species, which exhibit a positive shift for ZnPor-RuCuDAC and ZnPor-Ru₂DAC for an increased electron density of RuN₃ cores. However, as compared to RuN₃ monomer, ZnPor-RuCuDAC and ZnPor-Ru₂DAC show new binding energy peaks at 459.13/482.39 and 459.10/482.28 eV, respectively, which can be ascribable to the formation of Ru-Cu (ZnPor-RuCuDAC) or Ru-Ru (ZnPor-Ru₂DAC) diatomic pairs.

Similarly, CuN₃ present BE peaks of Cu(II) 2p_{3/2}/2p_{1/2} at 935.15/954.67 eV for Cu-N bond

(Supplementary Fig. 13d), which show a positive shift for ZnPor-RuCuDAC and ZnPor-Cu₂DAC for an increased electron density of CuN₃ cores. Contrastingly, as compared to CuN₃ monomer, ZnPor-RuCuDAC and ZnPor-Cu₂DAC show new binding energy peaks at 932.61/952.51 and 932.68/952.62 eV, respectively, which can be ascribable to the formation of Ru-Cu (ZnPor-RuCuDAC) or Cu-Cu (ZnPor-Cu₂DAC) diatomic pairs.

Therefore, as confirmed by EXAFS characterization, we believe **the above fitted XPS spectrum further indicate that compared to the single-atom Zn distribution, the Ru-Cu, Ru-Ru or Cu-Cu diatomic pairs are anchored on the COF structures via diatomic coordination assemblies of metal-N₃ connected by metal-metal bonding bridges.**

According to this comment, the manuscript (Page 9, the 2nd paragraph) is revised as: “Further, both ZnPor-RuCuDAC (Fig. 2k,l) and ZnPor-Cu₂DAC (Fig. 2m,n) exhibit Cu-N and Cu-metal shells, unlike Cu-based reference samples (Supplementary Fig. 12). **Also, as compared to ZnPor and RuN₃/CuN₃ monomers, all COFs present Zn(II) 2p_{3/2}/2p_{1/2} binding energy (BE) peaks ascribable to Zn-N bond (Supplementary Fig. 13), but emerge new BE peaks of Ru-Cu (ZnPor-RuCuDAC), Ru-Ru (ZnPor-Ru₂DAC) and Cu-Cu (ZnPor-Cu₂DAC) diatomic pairs besides Ru-N and/or Cu-N bond.¹⁵”**

The related XPS analysis part (Page 25-26) in supporting information is revised as: “The survey XPS (Supplementary Fig. 13a) indicate that Zn, Ru, Cu, C and N coexist in the precursors and/or these COFs. The high-resolution Zn2p XPS spectra (Supplementary Fig. 13b) show that ZnPor have Zn(II) 2p_{3/2}/2p_{1/2} binding energy (BE) peaks of 1021.81/1044.95 for Zn-N bond, which show a positive shift for ZnPor-RuCuDAC (~0.18 eV), ZnPor-Ru₂DAC (~0.22 eV) and ZnPor-Cu₂DAC (~0.23 eV), ascribable to a decreased electron density of the ZnPor cores. The high-resolution Ru3p XPS spectrum (Supplementary Fig. 13c) of RuN₃ exhibits two BE peaks at 462.57/485.83 and 464.43/488.93 eV for Ru-N and Ru-Cl bond, belonging to the Ru(II) species, which exhibit a positive shift for ZnPor-RuCuDAC and ZnPor-Ru₂DAC for an increased electron density of RuN₃ cores. However, as compared to RuN₃ monomer, ZnPor-RuCuDAC and ZnPor-Ru₂DAC show new binding energy peaks at 459.13/482.39 and 459.10/482.28 eV, respectively, which can be ascribable to the formation of Ru-Cu (ZnPor-RuCuDAC) or Ru-Ru (ZnPor-Ru₂DAC) diatomic pairs.

Similarly, CuN₃ monomer present BE peaks of Cu(II) 2p_{3/2}/2p_{1/2} for Cu-N and Ru-Cl bond

(Supplementary Fig. 13d), which show a positive shift for ZnPor-RuCuDAC and ZnPor-Cu₂DAC for an increased electron density of CuN₃ cores. Contrastingly, as compared to CuN₃ monomer, ZnPor-RuCuDAC and ZnPor-Cu₂DAC show new BE peaks at 932.61/952.51 and 932.68/952.62 eV, respectively, which can be ascribable to the formation of Ru-Cu (ZnPor-RuCuDAC) or Cu-Cu (ZnPor-Cu₂DAC) diatomic pairs. The above fitted XPS spectrum further indicate that compared to the single-atom Zn distribution, the Ru-Cu, Ru-Ru or Cu-Cu diatomic pairs are formed on the COF skeletons. Furthermore, the decreased electron density of ZnPor cores and increased one of RuN₃/CuN₃ can be ascribable to the intensive metal-to-metal charge transfer behaviors from ZnBPP to RuN₃/CuN₃ cores, demonstrating that energy transfer and electron diffusion occurred among the hetero-trimetallic cores in the extended two-dimensional networks of these COFs.”

Supplementary Figure 13. Survey (a), high-resolution Zn2p (b), Ru3p (c) and Cu2p (d) XPS spectra of ZnPor-RuCuDAC, ZnPor-Ru₂DAC, ZnPor-Cu₂DAC and their monomers, respectively.

2. In the EXAFS data with fitting, could the authors observe the Cu-N, Ru-N and so on bonds using any characterization methods? What was the actual structure of the Cu Ru, Zn with N? Besides, it was very strange that the XAS peaks were very sharp, although the EXAFS were ambiguous to analyze.

Re: We thank the referee for useful comment.

As seen in the revision of 1st comment, we analyze the chemical microenvironment structure of the monomers (ZnPor, RuN₃, CuN₃) and COFs (ZnPor-RuCuDAC, ZnPor-Ru₂DAC, ZnPor-Cu₂DAC) by high-resolution X-ray photoelectron spectra. ZnPor is observed the Zn-N bond, RuN₃ exhibits two BE peaks at 462.57 (Ru3p_{3/2}) and 485.83 (Ru3p_{1/2}) eV for Ru-N bond, and CuN₃ present BE peaks of Cu-N bond at 935.15 (Cu2p_{3/2}) and 954.67 eV (Cu2p_{1/2}). More importantly, besides these metal-N bonds, new binding energy peaks are observed in the synthesized COFs as compared to the monomers, which can be attributed to the Ru-Cu (ZnPor-RuCuDAC), Ru-Ru (ZnPor-Ru₂DAC) and Cu-Cu (ZnPor-Cu₂DAC) diatomic bonds.

Furthermore, according to the Fourier transformed radial distribution functions of the EXAFS spectra of monomers and these COFs with fitting results, the first-shell Zn K-edge peak (1.44 Å) of ZnPor is close to the ZnPc peak (1.45 Å) for the Zn-N bond (Supplementary Table 7); the central Zn²⁺ are coordinated with 4.0N, 4.1N, 3.9N and 4.0N atom at the ZnPor, ZnPor-RuCuDAC, ZnPor-Ru₂DAC and ZnPor-Cu₂DAC, respectively, in agreement with the reference (refers to: Nat. Commun. 2021, 12, 1354; J. Am. Chem. Soc. 2022, 144, 21328), and no additional Zn-Zn atomic scattering as referred to Zn foil (at 2.66 Å) is detected because of the single-atom site of the Zn species. Similarly, RuN₃ show coordination peaks of Ru-N (1.43 Å) with 3.1N atom, and CuN₃ have Cu-N shell scatterers with 3.0N coordination. By contrast, the predominant peak of Ru centers of ZnPor-RuCuDAC and ZnPor-Ru₂DAC exhibit 3.1N and 6.0N coordination of Ru-N path scattering (Supplementary Table 8), respectively; Cu centers of ZnPor-RuCuDAC and ZnPor-Cu₂DAC show Cu-N path scattering with 3.1N and 5.9N coordination number (Supplementary Table 9), respectively. Particularly, besides the coordination with the central N atoms, the Ru and/or Cu centers of these COFs emerge with 1.0 coordination number of Ru-Cu (ZnPor-RuCuDAC), Ru-Ru (ZnPor-Ru₂DAC), and Cu-Cu (ZnPor-Cu₂DAC) bonds.

Also, as indicated by the inductively coupled plasma-atomic emission spectrometry

(Supplementary Table 1), the sharp XAS peaks should be derived from the relatively high weight percentage of Zn, Ru and Cu elements in the COFs. **According to this comment, to clearly present the actual chemical microenvironment structure of the Zn, Ru, Cu with N, the EXAFS analysis part (Page 8-9) is revised as:** “ ...According to the Fourier transformed radial distribution functions of the EXAFS spectra of ZnPor and these COFs (Fig. 2d and Supplementary Table 7), the first-shell Zn K-edge peak (1.44 Å) is close to the ZnPc peak (1.45 Å), **coordinating with the central 4.0N (Zn–N₄) shell.** By contrast, no additional Zn–Zn atomic scattering as referred to Zn foil (at 2.66 Å) is detected because of the single-atom site of the Zn species, **with 4.1N, 3.9N and 4.0N atom at the ZnPor-RuCuDAC, ZnPor-Ru₂DAC and ZnPor-Cu₂DAC COFs.** Moreover, compared to the Ru foil and RuCl₃ reference (Fig. 2e and Supplementary Table 8), the *R* space Ru K-edge spectra of RuN₃ show coordination peaks of Ru–N (1.43 Å) and Ru–Cl (1.99 Å), but no Ru–Ru (2.76) peaks. Similarly, in the case of CuN₃, the *R*-space Fourier transformed Cu K-edge spectrum (Fig. 2f and Supplementary Table 9) exhibits peaks at 1.47 and 2.07 Å, corresponding to the Cu–N and Cu–Cl shell scatterers, respectively, and no additional Cu–Cu interaction referring to the Cu foil (at 2.61 Å) is observed. However, the Ru–Cl and/or Cu–Cl signals disappear in the spectra of the ZnPor-RuCuDAC, ZnPor-Ru₂DAC, and ZnPor-Cu₂DAC backbones, because of the Cl[–] disassociation from the Ru and/or Cu sites during the condensation of the ZnPor, RuN₃, and CuN₃ monomers, **indicating no Cl impurity on the COFs in consistent with element analysis.**

Comparatively, the predominant peak of Ru centers of ZnPor-RuCuDAC and ZnPor-Ru₂DAC (Fig. 2e) **exhibit 3.1N and 6.0N coordination of Ru–N path scattering, which is almost like** that of RuN₃ for nearly identical atom coordination environment. However, new peaks are observed at 2.70 Å (ZnPor-RuCuDAC) and 2.68 Å (ZnPor-Ru₂DAC),¹⁵ suggesting the presence of Ru–metal (Ru–Cu, Ru–Ru) diatomic configuration **with 1.0 atom coordination.** On the other hand, if we exclude the 3.1N and 5.9N Cu–N (1.47 Å) coordination in the ZnPor-RuCuDAC and ZnPor-Cu₂DAC (Fig. 2f), then the emerging Cu–metal peaks (**1.0 atom coordination**) that originate from the Cu foil reference indicate the formation of Cu–metal (Cu–Ru, Cu–Cu) dual-atom sites.¹⁷ According to the EXAFS fitting results (Supplementary Table 8,9), the Ru and/or Cu centers emerge with a coordination number of 1.0 for the Ru–Cu (ZnPor-RuCuDAC), Ru–Ru (ZnPor-Ru₂DAC), and Cu–Cu (ZnPor-Cu₂DAC) bonds in addition to the coordination with

the central N atoms. Wavelet transformation of the k_3 -weighted EXAFS was also conducted to identify the metal–N and metal–metal paths. The Zn centers in these three COFs merely show intensity maxima ($R+\Delta R = 1.43 \text{ \AA}$) for the Zn–N path (Supplementary Fig. 11). However, the Ru centers of ZnPor-RuCuDAC (Fig. 2g,h) and ZnPor-Ru₂DAC (Fig. 2i,j) show maximum intensity for the Ru–N and Ru–metal paths, compared to RuN₃ (Supplementary Fig. 12a). Further, both ZnPor-RuCuDAC (Fig. 2k,l) and ZnPor-Cu₂DAC (Fig. 2m,n) exhibit Cu–N and Cu–metal shells, unlike CuN₃ reference (Supplementary Fig. 12b). Also, as compared to ZnPor and RuN₃/CuN₃ monomers, all COFs present Zn(II) 2p_{3/2}/2p_{1/2} binding energy (BE) peaks ascribable to Zn–N bond (Supplementary Fig. 13), but emerge new BE peaks of Ru–Cu (ZnPor-RuCuDAC), Ru–Ru (ZnPor-Ru₂DAC) and Cu–Cu (ZnPor-Cu₂DAC) diatomic pairs besides Ru–N and/or Cu–N bond.¹⁵ These results collectively indicate that compared to the single-atom Zn distribution, ZnPor-RuCuDAC mainly exist Ru–Cu diatomic pairs via diatomic coordination assemblies of metal–N₃ connected by Ru–Cu bonding bridges, while ZnPor-Ru₂DAC and ZnPor-Cu₂DAC are anchored the Ru–Ru and Cu–Cu sites.”

Supplementary Table 7. EXAFS curve-fitting results for the structural parameters around Zn atom of various sample.

Sample	Shell	N	R (Å)	$\sigma^2 (\times 10^{-3} \text{ \AA}^2)$	ΔE_0 (eV)	R factor
Zn foil	Zn–Zn	12	2.66	1.5	5.8	0.083
Zn Pc	Zn–N	4.1	1.45	1.7	2.5	0.076
ZnPor	Zn–N	4.0	1.44	2.1	3.6	0.009
ZnPor-RuCuDAC	Zn–N	4.1	1.43	2.7	2.7	0.013
ZnPor-Ru ₂ DAC	Zn–N	3.9	1.43	3.1	3.1	0.019
ZnPor-Cu ₂ DAC	Zn–N	4.0	1.43	4.3	1.9	0.027

Shell: Coordination atom; N: Coordination number; R: Bond length; σ^2 : Debye-Waller factor; ΔE_0 : Inner potential correction; R factor: Goodness of fit. The fit was optimized in R space with a k-weight of 3. S_0^2 of Co–N path was set to 0.96 according to the fitting for experimental EXAFS of ZnPc.

Supplementary Table 8. EXAFS curve-fitting results for the structural parameters around Ru atom of various sample.

Sample	Shell	N	R (Å)	σ^2 ($\times 10^{-3}$ Å)	ΔE_0 (eV)	R factor
Ru foil	Ru–Ru	12.0	2.76	4.5	7.3	0.007
RuCl ₃	Ru–Cl	6.1	1.98	3.8	6.8	0.138
RuN ₃	Ru–N	3.1	1.43	2.2	5.4	0.009
	Ru–Cl	3.0	1.99	2.5	5.4	0.009
ZnPor-RuCuDAC	Ru–N	3.1	1.43	3.2	3.8	0.037
	Ru–Cu	1.0	2.70	3.4	3.8	0.037
ZnPor-Ru ₂ DAC	Ru–N	6.0	1.43	2.9	4.6	0.018
	Ru–Ru	1.0	2.68	3.1	4.6	0.018

Shell: Coordination atom; *N*: Coordination number; *R*: Bond length; σ^2 : Debye-Waller factor; ΔE_0 : Inner potential correction; *R* factor: Goodness of fit. The fit was optimized in *R* space with a *k*-weight of 3. S_0^2 of Ru-N path was set to 0.92 according to the fitting for experimental EXAFS of RuCl₃.

Supplementary Table 9. EXAFS curve-fitting results for the structural parameters around Cu atom of various sample.

Sample	Shell	N	R (Å)	σ^2 ($\times 10^{-3}$ Å)	ΔE_0 (eV)	R factor
Cu foil	Cu–Cu	12.0	2.68	3.2	2.4	0.017
CuPc	Cu–Cl	4.0	1.48	3.8	3.5	0.072
CuN ₃	Cu–N	3.0	1.47	1.7	3.2	0.006
	Cu–Cl	3.0	2.07	2.1	3.2	0.006
ZnPor-RuCuDAC	Cu–N	3.1	1.47	1.6	3.6	0.016
	Ru–Cu	1	2.61	1.8	3.6	0.016
ZnPor-Cu ₂ DAC	Cu–N	5.9	1.47	2.6	1.9	0.027
	Cu–Cu	1.0	2.59	2.9	1.9	0.027

Shell: Coordination atom; *N*: Coordination number; *R*: Bond length; σ^2 : Debye-Waller factor; ΔE_0 : Inner potential correction; *R* factor: Goodness of fit. The fit was optimized in *R* space with a *k*-weight of 3. S_0^2 of Cu-N path was set to 0.95 according to the fitting for experimental EXAFS of CuPc.

3. For the photocatalytic CO₂ reduction process, the transfer and life of the excited electrons can be the time-resolved PL and fs-TA absorption spectra, which can obtain the migration mechanism of photogenerated charge-holes in ZnPor-RuCuDAC matrices during the CO₂ reduction process.

Re: We appreciate the comment from the constructive referee.

To confirm the charge-transfer mechanism between the ZnPor unit and dual-atom active sites in the ZnPor-RuCuDAC matrices during the CO₂ reduction process, we have conducted the characterizations of picosecond transient absorption spectroscopy and time-resolved photoluminescence on the revised manuscript.

Accordingly, the relative part (Page 13, the 2nd paragraph) is revised as: “To comprehensively reveal the charge-transfer mechanism between the ZnPor unit and dual-atom active sites, the XAS of metal L-edge was studied since it can directly determine whether the illumination causes a change in the electronic structure of metal centers at the COFs.¹⁵ Under different irradiation times, the Zn L₃-edge of ZnPor-RuCuDAC shifts toward higher photon energies (Supplementary Fig. 29), whereas the Ru and Cu L₃-edges move to the lower photon-energy side (Supplementary Fig. 30). The picosecond transient absorption (Fs-TA) spectroscopy of ZnPor-RuCuDAC (Supplementary Fig. 31) exhibit the broader and more intensive transient absorption band at 552–736 nm than the metal-deficient COF counterparts (ZnPor-N₃ COF and H₂Por-RuCuDAC), ascribable to the charge transfer between ZnPor (Por) cores and Ru–Cu diatomic sites. The time-resolved photoluminescence (TRPL) of ZnPor-RuCuDAC (Supplementary Fig. 32, Supplementary Table 12) show the fastest decay lifetime than ZnPor-N₃ COF and H₂Por-RuCuDAC, especially the τ_2 lifetime with 49.6% percentage decay, with the shortest average decay lifetime (1.53 ns). These observations suggest that the Ru/Cu centers can accept the photogenerated electrons of ZnPor units and act as the dual-atom active sites (Supplementary Fig. 16b),^{40,42} leading to the formation of low-valence Ru/Cu species during the illumination.”

The related analysis part of picosecond transient absorption spectroscopy and time-resolved photoluminescence in supporting information is revised as: “For comparison, metal-deficient COF counterparts of ZnPor-N₃ COF (Ru- and Cu-free) and H₂Por-RuCuDAC (Zn-free) were synthesized for reaching a comprehensive understanding of the charge-transfer

mechanism between the ZnPor unit and dual-atom active sites, and the corresponding picosecond transient absorption (Fs-TA) spectroscopy was performed (Supplementary Fig. 31). After being excited by a pump pulse with a wavelength of 400 nm, the TA spectra of ZnPor-N₃ COF showed a pronounced negative peak at ca. 542 nm, which is assigned to ground-state bleach (GSB) and reflects the excited state relaxation. Except for the GSB peak, an extra positive absorption band at ca. 553–684 nm was observed in the TA spectra of H₂Por-RuCuDAC COF at delay time, but it was not detected in the spectra of ZnPor-N₃ COF. Furthermore, ZnPor-RuCuDAC exhibit the much broader and more intensive absorption band at 552–736 nm. This finding demonstrates that the fluctuant TA of H₂Por-RuCuDAC and ZnPor-RuCuDAC is attributed to the charge transfer between ZnPor (Por) cores and Ru–Cu diatomic sites according to the energy transfer and electron diffusion procedure of MLCT, ILCT and LMCT. This conjecture can be further confirmed by picosecond TRPL spectra (Supplementary Fig. 32 and Supplementary Table 12) of these COFs, where the time decays are fitted with double exponential function ($\Delta A(t) = \Delta A_0 + A_1 e^{-t/\tau_1} + A_2 e^{-t/\tau_2}$), resulting in one component with a shorter lifetime (τ_1 , contributing radiative fluorescence quenching) and another component with a longer lifetime (τ_2 , reflecting nonradiative recombination). As seen, the ZnPor-RuCuDAC showed the fastest decay lifetime than ZnPor-N₃ COF and H₂Por-RuCuDAC, especially for τ_2 lifetime with 49.6% percentage decay. The average fluorescence lifetime (τ_{ave}) was calculated to be 3.15 (ZnPor-N₃ COF), 2.56 (H₂Por-RuCuDAC) and 1.53 (ZnPor-RuCuDAC) ns, respectively, implying more fluent photogenerated charge separation over ZnPor-RuCuDAC.”

Supplementary Figure 31. Fs-TA spectra of (a) ZnPor-N₃ COF, (b) H₂Por-RuCuDAC and (c) ZnPor-RuCuDAC obtained at 400 nm excitation at different pump-probe decay lifetimes.

Supplementary Figure 32. TRPL spectra of (a) ZnPor-N₃ COF, (b) H₂Por-RuCuDAC and (c) ZnPor-RuCuDAC.

ZnPor-RuCuDAC excited and detected by 400 and 561 nm, respectively.

Supplementary Table 12. Comparison of the fluorescence decay data of ZnPor-N₃ COF, H₂Por-RuCuDAC and ZnPor-RuCuDAC.

sample	ZnPor-N ₃ COF	H ₂ Por- RuCuDAC	ZnPor- RuCuDAC
τ_1 (ns)	0.83	0.72	0.63
τ_2 (ns)	6.73	5.86	3.39
τ_{ave} (ns)	3.15	2.56	1.53

4. The characterization of ZnPor-RuCuDAC is not enough to understand the material's state. The authors reported XANESs, and atomic-resolution STEM with only a few snapshots without citations. What were the chemical properties of ZnPor-RuCuDAC? Can the authors prepare more direct evidence for proving the Ru–Cu sites?

Re: We appreciate the referee bringing this important issue to our attention.

First, as seen in the point-by-point response of this reviewer 1st and 2nd comment, to present more information about the diatomic sites in these synthesized COFs, we have supplemented the analysis of chemical microenvironment structure of the monomers (ZnPor, RuN₃, CuN₃) and their COFs (ZnPor-RuCuDAC, ZnPor-Ru₂DAC, ZnPor-Cu₂DAC) by high-resolution X-ray photoelectron spectra. The fitted XPS spectrum further indicate that contrasted to the single-atom Zn distribution, the Ru-Cu, Ru-Ru or Cu-Cu diatomic pairs are anchored on the COF structures via diatomic coordination assemblies of metal–N₃ connected by metal–metal bonding bridges.

Second, to further characterize the state of these porphyrin-based diatomic COFs, we analyze the thermogravimetric curves of ZnPor-RuCuDAC, ZnPor-Ru₂DAC and ZnPor-Cu₂DAC in the air. From thermogravimetric analysis (Supplementary Fig. 8b–d), all three COFs are thermally stable up to 368 °C in the air. Combining the elemental analysis (C, H, N, Zn, Ru, Cu), FT-IR spectra, Solid-state ¹³C NMR and nitrogen sorption isotherms, the COFs

physicochemical property is further evaluated in the in the revision

Third, to evaluate the chemical property of these COFs, we have analyzed the PXRD (Supplementary Fig. 8a) pattern of ZnPor-RuCuDAC before and after treated in different acid/alkaline solvents and XPS (Supplementary Fig. 24) of the recovered COFs' sample after the 40 h photocatalytic reaction. As contrasted to the original PXRD pattern, ZnPor-RuCuDAC (Supplementary Fig. 8a) do not vary in the peak position and intensity under different solvents obviously, indicating highly chemical-stable crystallinity under acid/alkaline conditions. As seen in the survey XPS of recovered ZnPor-RuCuDAC, ZnPor-Ru₂DAC and ZnPor-Cu₂DAC after 40 h photoreaction, Zn, Ru, Cu, C and N still coexist in these COFs. The high-resolution Zn 2p XPS spectra exhibit two Zn2p_{3/2}/2p_{1/2} binding energy peaks which is like the raw materials, suggesting the Zn valence state of Zn was +2 at the recovered COFs. Similarly, the Ru 3p XPS spectrum could be deconvoluted into two group asymmetrical BE peaks of Ru²⁺ species for the Ru–N and Ru–metal bonds, and Cu 2p XPS spectrum could be fitted into two group asymmetrical BE peaks of Cu²⁺ species for the Cu–N and Cu–metal bonds, similar to the raw one. Moreover, the ZnPor-RuCuDAC and its recovered samples from the 40 h photoreaction exhibit quite similar DRS and PXRD. These results about chemical property indicate that the COFs bear good long-time stability for CO₂RR conversion without significant crystallinity destruction.

Supplementary Figure 8. (a) PXRD pattern of ZnPor-RuCuDAC before and after treated in different solvents for 40 h. In a typical experiment, 10 mg of ZnCu-COF was immersed in the solvents (*n*-hexane, 0.5 M HCl, 6.0 M NaOH) at room temperature. Thermogravimetric curves of (b) ZnPor-RuCuDAC, (c) ZnPor-Ru₂DAC and (d) ZnPor-Cu₂DAC in the air.

Supplementary Figure 24. Survey (a), high-resolution Zn2p (b), Ru3p (c) and Cu2p (d) XPS spectra of ZnPor-RuCuDAC, ZnPor-Ru₂DAC, ZnPor-Cu₂DAC after 40 h photoreaction, respectively.

Fourth, based on this suggestion, some valuable references are cited in the revision and listed in References. These references can support the atomic structure by XANES (ref. 14, 15, 17, 29, 40), and clearly observe the pore channels and individual building compartments through aberration-corrected scanning transmission electron techniques (ref. 42, 43) and scanning tunneling microscopy (ref. 44). Accordingly, the original references are also changed in both manuscript and reference list. **For the details, please refer to document named as “Highlighted revision of NCOMMS-23-00556”.**

14 Li, G. F. et al. Dinuclear metal complexes: multifunctional properties and applications. *Chem. Soc. Rev.* **49**, 765–838 (2020).

15 Wang, J. M. et al. Highly durable and fully dispersed cobalt diatomic site catalysts for CO₂ photoreduction to CH₄. *Angew. Chem. Int. Ed.* **61**, e202113044 (2022).

17 Liu, Q. et al. Regulating the *OCCHO intermediate pathway towards highly selective

photocatalytic CO₂ reduction to CH₃CHO over locally crystallized carbon nitride. *Energy Environ. Sci.* **15**, 225–233 (2022).

29 Wang, W. et al. Photocatalytic C-C coupling from carbon dioxide reduction on copper oxide with mixed-valence Copper(I)/Copper(II). *J. Am. Chem. Soc.* **143**, 2984–2993 (2021).

40 Wang, J. M. et al. Porphyrin conjugated polymer grafted onto BiVO₄ nanosheets for efficient Z-scheme overall water splitting via cascade charge transfer and single-atom catalytic sites. *Adv. Energy Mater.* **11**, 2003575 (2021).

42 Liu, K. et al. On-water surface synthesis of crystalline, few-layer two-dimensional polymers assisted by surfactant monolayers. *Nat. Chem.* **11**, 994–1000 (2019).

43 Joshi, T. et al. Local electronic structure of molecular heterojunctions in a single-layer 2D covalent organic framework. *Adv. Mater.* **31**, 1805941 (2019).

44 Liu, W. B. A scalable general synthetic approach toward ultrathin imine-linked two-dimensional covalent organic framework nanosheets for photocatalytic CO₂ reduction. *J. Am. Chem. Soc.* **141**, 17431–17440 (2019).

5. The authors need to provide detailed proposed reaction mechanism figure with proper band edge and band gap values to understand the charge transfer process and correlate with existed results.

Re: We thank the reviewer for the helpful comment.

With schematic diagram of band gap structure (Supplementary Fig. 16), the energy transfer and electron diffusion process over these diatomic COFs is presented in the revision. To discover the diatomic-type effect on the electronic band structure, synchrotron radiation photoemission spectroscopy (SRPES) was conducted on these COFs. The SRPES results indicate that their valence band (*VB*, vs. NHE) potentials are 1.05 (ZnPor-RuCuDAC), 0.95 (ZnPor-Ru₂DAC), and 0.93 (ZnPor-Cu₂DAC) eV (Supplementary Fig. 14a–c). The Kubelka–Munk method (Supplementary Fig. 15b–d) was employed to determine the optical bandgap energies (*E_g*), which were found to be 1.75 (ZnPor-RuCuDAC), 1.63 (ZnPor-Ru₂DAC), and 1.66 (ZnPor-Cu₂DAC) eV. Therefore, the relative energy band structure of these COFs (Supplementary Table 10) indicate that the conductive band potentials are appropriate for formation of typical photocatalytic products (Supplementary Fig. 16a and Supplementary Table

11), theoretically preferable for CO₂RR.

For further discussing the photogenerated electron process, the relative part in manuscript (Page 14) is revised as: “To comprehensively reveal the charge-transfer mechanism between the ZnPor unit and dual-atom active sites, the XAS of metal L-edge was studied since it can directly determine whether the illumination causes a change in the electronic structure of metal centers at the COFs.¹⁵ Under different irradiation times, the Zn L₃-edge of ZnPor-RuCuDAC shifts toward higher photon energies (Supplementary Fig. 29), whereas the Ru and Cu L₃-edges move to the lower photon-energy side (Supplementary Fig. 30). **The picosecond transient absorption (Fs-TA) spectroscopy of ZnPor-RuCuDAC (Supplementary Fig. 31) exhibit the broader and more intensive transient absorption band at 552–736 nm than the metal-deficient COF counterparts (ZnPor-N₃ COF and H₂Por-RuCuDAC), ascribable to the charge transfer between ZnPor (Por) cores and Ru–Cu diatomic sites. The time-resolved photoluminescence (TRPL) of ZnPor-RuCuDAC (Supplementary Fig. 32, Supplementary Table 12) show the fastest decay lifetime than ZnPor-N₃ COF and H₂Por-RuCuDAC, especially the τ_2 lifetime with 49.6% percentage decay, with the shortest average decay lifetime (1.53 ns). These observations suggest that the Ru/Cu centers can accept the photogenerated electrons of ZnPor units and act as the dual-atom active sites (Supplementary Fig. 16b),^{40,42} leading to the formation of low-valence Ru/Cu species during the illumination.”**

Also, the relative part in supporting information is revised as: “For comparison, metal-deficient COF counterparts of ZnPor-N₃ COF (Ru- and Cu-free) and H₂Por-RuCuDAC (Zn-free) were synthesized for reaching a comprehensive understanding of the charge-transfer mechanism between the ZnPor unit and dual-atom active sites, and the corresponding picosecond transient absorption (Fs-TA) spectroscopy was performed (Supplementary Fig. 31). After being excited by a pump pulse with a wavelength of 400 nm, the TA spectra of ZnPor-N₃ COF showed a pronounced negative peak at ca. 542 nm, which is assigned to ground-state bleach (GSB) and reflects the excited state relaxation. Except for the GSB peak, an extra positive absorption band at ca. 553–684 nm was observed in the TA spectra of H₂Por-RuCuDAC COF at delay time, but it was not detected in the spectra of ZnPor-N₃ COF. Furthermore, ZnPor-RuCuDAC exhibit the much broader and more intensive absorption band at 552–736 nm. This finding demonstrates that the fluctuant TA of H₂Por-RuCuDAC and

ZnPor-RuCuDAC is attributed to the charge transfer between ZnPor (Por) cores and Ru–Cu diatomic sites according to the energy transfer and electron diffusion procedure of MLCT, ILCT and LMCT. This conjecture can be further confirmed by picosecond TRPL spectra (Supplementary Fig. 32 and Supplementary Table 12) of these COFs, where the time decays are fitted with double exponential function ($\Delta A(t) = \Delta A_0 + A_1e^{-t/\tau_1} + A_2e^{-t/\tau_2}$), resulting in one component with a shorter lifetime (τ_1 , contributing radiative fluorescence quenching) and another component with a longer lifetime (τ_2 , reflecting nonradiative recombination). As seen, the ZnPor-RuCuDAC showed the fastest decay lifetime than ZnPor-N₃ COF and H₂Por-RuCuDAC, especially for τ_2 lifetime with 49.6% percentage decay. The average fluorescence lifetime (τ_{ave}) was calculated to be 3.15 (ZnPor-N₃ COF), 2.56 (H₂Por-RuCuDAC) and 1.53 (ZnPor-RuCuDAC) ns, respectively, implying more fluent photogenerated charge separation over ZnPor-RuCuDAC.”

Supplementary Figure 16. Schematic diagram of (a) energy band structures and (b) simplified model for electron diffusion within these PMD-like diatomic COFs. (GS = ground state, PL = photoluminescence, MLCT = metal-to-ligand charge transfer, ILCT = intra-ligand charge transfer, LMCT = ligand-to-metal charge transfer)

Reviewer #2 (Remarks to the Author): This work synthesized the diatomic-site COF catalyst, ZnPor-RuCuDAC, by bridging Zn-porphyrin and Ru/Cu-pincer complexes. Compared with the homoatomic counterparts ZnPor-Cu₂DAC and ZnPor-Ru₂DAC, it shows a best acetate selectivity in photocatalytic CO₂ reduction reaction. The author believes that this is due to the distinct charge distributions of heteronuclear Ru-Cu sites, which suppress the electrostatic repulsion between the two neighboring *CO intermediates, resulting in a weak dipole-dipole interaction and promoting the C-C coupling process. The production of acetate was deeply analyzed by many in-situ experiments and calculations. However, there are some problems in the analysis of the ZnPor-RuCuDAC structure, including the framework structure of COF and the local structure of active sites. This may affect the establishment of the model in the subsequent calculation, resulting in the deviation of the calculation results.

1. How can the authors get the specific ZnPor-RuCuDAC COF structure in Supplementary Scheme 1? If it is only a conjecture or schematic diagram, whether the description in the element analysis results “the experimental weight percentage of elemental C, H, N, Zn, Ru and Cu are closed to their respective theoretical compositions” is not logical, because the author has not clearly given the skeleton structure here.

Re: Many thanks for bringing this important issue to our attention.

We are very sorry for our careless descriptions on the calculated process of theoretical elemental values (C, H, N, Zn, Ru and Cu). As depicted in the Experimental Section, this porphyrin-based COFs (ZnPor-RuCuDAC, ZnPor-Ru₂DAC and ZnPor-Cu₂DAC) were synthesized by Sonogashira coupling reaction (condensation of aldehyde and amino group), which the stoichiometric ratio of precursor monomers (ZnPor, RuN₃ and CuN₃) will be tightly controlled when this coupling reaction terminate. ZnPor monomer in the junction with eight aldehyde substituents are especially easier to condense with amino group of RuN₃/CuN₃ monomers through Sonogashira coupling reaction (please refer to: Adv. Funct. Mater. 2021, 31, 2107290; ACS Energy Lett. 2018, 3, 2544-2549; Angew. Chem. Int. Ed. 2016, 55, 15712-15727; J. Mater. Chem. A, 2018, 6, 8349-8357; ACS Appl. Mater. Interfaces 2019, 11, 11466-11473; J. Mater. Chem. A, 2019, 7, 3112-3119). On the other hand, the ultimate COF products are collected by filtration and the precipitate was washed with THF, CHCl₃, H₂O and CH₃OH, then rigorously extracted by Soxhlet for 24 h with THF, CHCl₃, H₂O and CH₃OH, respectively. This

process can basically remove the oligomer products. Also, those COFs exhibit poor solubility in all kinds of solvents, indicating its macromolecular structure. Therefore, based on this consideration, the ZnPor-RuCuDAC COF structure in Supplementary Scheme 1 is an engineered structure, and its reasonability has been confirmed by a series of characterization (FTIR spectra, Solid-state ^{13}C NMR, Pawley-refined PXRD, XPS, STEM, XAS, et al) in the revised manuscript.

After carefully referred to some published papers on porphyrin-based polymers or COFs for energy conversion (please refer to: *Adv. Mater.* 2019, 31, 1805941-1805946; *Angew. Chem. Int. Ed.* 2019, 131, 1-7; *J. Am. Chem. Soc.* 2018, 140, 1116-1122; *Chem. Commun.* 2017, 53, 4461-4464; *Chem. Commun.* 2019, 55, 1627-1630; *ACS Applied Materials & Interfaces* 2019, 11, 1520-1528), we think cyclic structure (Supplementary Scheme 1) of those porphyrin-based photocatalysts possibly present its real structure better.

However, the theoretical values of elemental of C, H, N, Zn, Ru and Cu is not just obtained by the Supplementary Scheme 1 structure. According to the Sonogashira coupling reaction, it will take off 8eq of H_2O and 4eq of Cl if ZnPor is 1eq in this process. Noteworthily, due to the possible exists of few terminal groups such as -CHO and -Cl in these COFs (see references: *ACS Energy Lett.* 2018, 3, 2544-2549; *J. Am. Chem. Soc.* 2018, 140, 1116-1122; *Angew. Chem. Int. Ed.* 2019, 131, 1-7), the synthesis by coupling reaction cannot be reacted very completely for the polymer, which may lead to the slight difference of experimental and theoretical values. Factually, as presented in Supplementary Table 1, the theoretical elemental values were calculated by using the following formula: 1) ZnPor-RuCuDAC [$1 \times (\text{ZnPor}) + 2 \times (\text{RuN}_3) + 2 \times (\text{CuN}_3) - 8 \times (\text{H}_2\text{O}) - 4 \times (\text{Cl})$]/5 = $\text{C}_{25.6}\text{H}_{14.4}\text{N}_{6.4}\text{Zn}_{0.2}\text{Ru}_{0.4}\text{Cu}_{0.4}$; 2) ZnPor-Ru₂DAC [$1 \times (\text{ZnPor}) + 4 \times (\text{RuN}_3) - 8 \times (\text{H}_2\text{O}) - 4 \times (\text{Cl})$]/5 = $\text{C}_{25.6}\text{H}_{14.4}\text{N}_{6.4}\text{Zn}_{0.2}\text{Ru}_{0.8}$; 3) ZnPor-Cu₂DAC [$1 \times (\text{ZnPor}) + 4 \times (\text{RuN}_3) - 8 \times (\text{H}_2\text{O}) - 4 \times (\text{Cl})$]/5 = $\text{C}_{25.6}\text{H}_{14.4}\text{N}_{6.4}\text{Zn}_{0.2}\text{Cu}_{0.8}$. This assessment method is often used in the COFs' materials (please refer to: *J. Am. Chem. Soc.* 2019, 141, 17431-17440; *Adv. Energy Mater.* 2021, 11, 2003575; *J. Am. Chem. Soc.* 2018, 140, 16124-16133).

To avoid the misunderstanding of theoretical elemental values of these COFs, the related part in the in supporting information is revised as following:

Supplementary Table 1. Contents of Zn, Ru and Cu elements in Zn-COF, Cu-COF and ZnCu-COF determined by ICP-OES.

COF	C(%) ^{a)}	H(%) ^{a)}	N(%) ^{a)}	Zn(%) ^{b)}	Ru(%) ^{b)}	Cu(%) ^{b)}
ZnPor-RuCuDAC ^{c)}	62.72 (62.67)	3.02 (2.96)	18.16 (18.28)	2.62 (2.67)	8.29 (8.24)	5.20 (5.18)
ZnPor-Ru ₂ DAC ^{c)}	60.76 (60.81)	2.83 (2.87)	17.60 (17.73)	2.63 (2.59)	16.18 (16.00)	0 (0)
ZnPor-Cu ₂ DAC ^{c)}	64.70 (64.65)	3.13 (3.05)	18.73 (18.85)	2.71 (2.76)	0 (0)	10.73 (10.69)

^{a)} Data calculated from element analysis results.

^{b)} Data determined with ICP-OES.

^{c)} Data in parentheses were theoretically values calculated according to the following formula.

Note: The stoichiometric ratio of precursor monomers (ZnPor, RuN₃ and CuN₃) will be tightly controlled when the COF coupling reaction terminate. Because it will take off 8eq of H₂O and 4eq of Cl if ZnPor is 1eq in this process, the theoretical elemental values were calculated by using the following formula: 1) ZnPor-RuCuDAC [1×(ZnPor) + 2×(RuN₃) + 2×(CuN₃) – 8×(H₂O) – 4×Cl]/5 = C_{25.6}H_{14.4}N_{6.4}Zn_{0.2}Ru_{0.4}Cu_{0.4}; 2) ZnPor-Ru₂DAC [1×(ZnPor) + 4×(RuN₃) – 8×(H₂O) – 4×Cl]/5 = C_{25.6}H_{14.4}N_{6.4}Zn_{0.2}Ru_{0.8}; 3) ZnPor-Cu₂DAC [1×(ZnPor) + 4×(RuN₃) – 8×(H₂O) – 4×Cl]/5 = C_{25.6}H_{14.4}N_{6.4}Zn_{0.2}Cu_{0.8}.

2. *The name of the corresponding catalysts should be clearly marked in Figure 1b-d.*

Re: We thank the referee for useful comment.

This suggestion does help us to enhance the article readability. Accordingly, Figure 1b–d has been clarified in the revised manuscript.

3. *The spatial atomic cross-distribution of Zn, Ru and Cu elements on the ZnPor-RuCuDAC substrate in Figures 1g-j cannot match the AB-staggered stacking model in Supplementary Figure 4. In the AB-staggered stacking model in Supplementary Figure 4, the Zn element and Ru/Cu elements cannot appear at the same position when viewed from a vertical perspective. The inset in Figure 1g also shows the same information. This means that there may be problems with the skeleton structure of COF given in the article.*

Re: We appreciate the comment from the constructive referee.

We are very sorry for our unclear presentation on AB-staggered stacking model at Fig. 1g.

To better understand the AB-staggered stacking model, a replacement of the simulated AB-staggered structure is added in the Fig. 1g, where the A-layer is labeled as blue color and B-layer is labeled as yellow color. As seen, by AC-ADF-STEM with an emerging differential phase contrast technique at an extremely low beam current, the well-defined periodic structure of pore channels and individual building compartments can be obtained, matching well with the simulated AB-staggered stacking structure of ZnPor-RuCuDAC. On the other hand, according to the nonlocal density functional theory, ZnPor-RuCuDAC with AA-eclipsed, AB-staggered, slipped ABC-1, slipped ABC-2, and slipped ABC-3 stacking models show pore size distributions centered at 3.41, 1.67, 1.25, 1.25, and 0.93 nm, respectively. However, the measured unit parameters of ZnPor-RuCuDAC COF are measured with $a = b = 1.67 \pm 0.1$ nm ($\alpha = 90 \pm 0.3^\circ$), in good agreement with the theoretical AB-staggered pore size.

Also, Figure 1g-j just confirm that the Zn, Ru and Cu atoms are uniformly distributed on the ZnPor-RuCuDAC backbone, which cannot directly confirm whether the Zn and Ru/Cu elements are the same spatial position or not. To clearly distinguish the spatial position of Zn, Ru and Cu elements, the relative atom spatial coordinate (Zn, Ru, Cu), obtained from the 3D electron diffraction tomography of DPC image (Nature 2020, 586, 549-554), is analyzed in the revised manuscript (please refer to the Supplementary Fig. 10). Ru/Cu atoms are indeed the same spatial position, and the Zn and Ru/Cu atoms show a 90° phase difference on the spatial location, further confirming the successful fabrication of AB-staggered stacking ZnPor-RuCuDAC COF with heteronuclear Ru-Cu diatomic sites.

Moreover, in order to elucidate the structures of these COFs and unit cell parameters, three types of possible 2D structures have been generated for ZnPor-RuCuDAC: AA-eclipsed (Supplementary Fig. 4a), AB-staggered (Supplementary Fig. 4b), and slipped ABC-staggered (Supplementary Fig. 4c–e) stacking models. The simulated powder X-ray diffraction (PXRD) analyses (Supplementary Fig. 4f) reveal that the AA-eclipsed ($x=0, y=0$) stacking shows the first intense peak at a low angle of 2.46° , corresponding to the (110) reflection; also, it shows minor peaks at 3.39° and 4.92° for the (202) and (004) reflection planes, respectively. For the AB-staggered ($x=1, y=1$) stacking, the most intense peak corresponding to the (200) reflection plane appears at $\sim 4.41^\circ$, along with other minor peaks at 3.06° and 9.57° that originate from the (202) and (211) reflection planes, respectively. The slipped ABC-staggered stacking exhibits

the same PXRD peaks at 2.46°, 3.39°, 4.81°, 6.62°, 7.08°, 7.89°, 13.25°, and 18.49°, corresponding to the (200), (202), (004), (210), (212), (014), (420), and (031) planes, respectively. However, both the ABC-1 ($x = 0.5, y = 0$) and ABC-2 ($x = 0, y = 0.5$) stacking models show the most intense peak at 4.81°, whereas the slipped ABC-3 ($x = 0.5, y = 0.5$) model shows the strongest peak intensity at 3.39°. The experimental PXRD profiles of ZnPor-RuCuDAC, ZnPor-Ru₂DAC, and ZnPor-Cu₂DAC exhibit three main diffraction peaks at 3.06°, 4.42°, and 9.57°, which are consistent with those obtained in the simulated diffraction pattern of AB-staggered stacking model. This result suggests that these diatomic COFs adopt an AB-stacked structure, which is evidenced by the negligible difference obtained in the Pawley refinement results.

Further, the density functional tight binding method with Lennard–Jones function was utilized to evaluate crystal stacking energies of ZnPor-RuCuDAC quantitatively (Supplementary Table 5). Evidently, the AA-eclipsed, AB-staggered, slipped ABC-1, slipped ABC-2, and slipped ABC-3 stacking exhibit total per-layer crystal stacking energies of 57.82, 134.53, 97.25, 97.25, and 118.79 kcal mol⁻¹, respectively, indicating that AB-staggered stacking is more favorable than the other four isomeric structures.

Therefore, based on the above consideration, we think AB-staggered stacking structure of these COF photocatalysts possibly present its real structure better than the other models if this reviewer agree.

Supplementary Figure 10. The relative Zn, Ru and Cu atom spatial location in ZnPor-RuCuDAC-COF backbone, obtained from 3D electron diffraction tomography of DPC images.

Accordingly, the corresponding part of manuscript (Page 8) was revised as:

“Interestingly, aberration-corrected annular dark-field STEM (AC-ADF-STEM), with an emerging differential phase contrast (DPC) technique at an extremely low beam current, was applied to generate an integrated DPC image over the atom distribution in the COF backbone.⁶ The parameters of individual building compartments are measured with $a = b = 1.67 \pm 0.1$ nm ($\alpha = 90 \pm 0.3^\circ$) in ZnPor-RuCuDAC COF, in good agreement with the AB-staggered pore size of simulated structure (inserted at Fig. 1g and Supplementary Fig. 4b). The 3D electron diffraction tomography of DPC images (Fig. 1h–j) unveil the equably atomic cross-distribution of Zn, Ru and Cu elements on the ZnPor-RuCuDAC substrate, where the Zn and Ru/Cu atoms show a 90° phase difference on the spatial location (Supplementary Fig. 10).”

4. Page 6, “The elemental analysis results of ZnPor-RuCuDAC single crystal (Supplementary Fig. 8c-f) show a uniform distribution of the Zn, Ru, and Cu elements on the scaffolds.” Note that this material is not a single crystal.

Re: We thank the reviewer for the professional comment.

Corresponding, we have modified the description in the revision.

5. Does “the rising edges” means edge energy? It is suggested that the authors mark the edge positions as “edge energy” or “pre-edge energy” in Supplementary Table 6 for easy identification. In addition, the representation of Figures 2a-c cannot effectively identify the difference between different catalysts. Is the author considering a different form of expression?

Re: Thanks a lot for useful comment.

Accordingly, the title of Supplementary Table 6 is corrected, and the description of “the rising edges” has been modified in the revision. Also, Figure 2a–c is re-drawn in the revised manuscript. Please refer to the revised manuscript for the detailed modification information.

6. The Zn-Zn peak in Figure 2d is labeled incorrectly.

Re: We appreciate the reviewer thorough assessment.

Accordingly, Figure 2d has been corrected in the revised manuscript.

7. The wavelet transform provides contributions of near neighbors in both R-space and k-space, and its abscissa is related to the atomic weight. Atoms with small atomic numbers have weak scattering ability to photoelectrons, and their strongest oscillation will occur in the lower k part, while the opposite is true for atoms with large atomic numbers. However, the comparison between Figure 2g and Figure 2i shows that Ru with a larger atomic number corresponds to a smaller k value. This is not reasonable. The same problems apply to the analysis of ZnPor-RuCuDAC and ZnPor-Cu2DAC, as well as a large number of the wavelet transform analyses in Supplementary Information. In addition, Ru-N in ZnPor-RuCuDAC and ZnPor-Ru2DAC or Cu-N in ZnPor-RuCuDAC and ZnPor-Cu2DAC should have extremely approximate values. This part of EXAFS data needs more accurate analysis.

Re: Thanks a lot for this professional comment!

We are sorry for our carelessness on wavelet transforms of the k^3 -weighted EXAFS profiles. Because R-space data are used rather than k-space data under wavelet transform, there is confused about wavelet transform results. Now, we have correctly used the k^3 -weighted EXAFS data to make the wavelet transform analysis in the revised manuscript, and all confused profiles are replaced. We are very grateful for this Reviewer's useful and friendly reminder on this issue, and will focus it to avoid such worries in our future researches.

Fig. 2 (a) Zn, (b) Ru, and (c) Cu K-edge XANESs (top layer), their first derivatives (bottom layer), and the Fourier transformation results of the (d) Zn, (e) Ru and (f) Cu K-edge EXAFSs of the diatomic COFs. Wavelet transforms of the k_3 -weighted EXAFS profiles of Ru in (g,h) ZnPor-RuCuDAC and (i,j) ZnPor-Ru₂DAC and those of Cu in (k,l) ZnPor-RuCuDAC and (m,n) ZnPor-Cu₂DAC.

Supplementary Figure 11. Wavelet transforms of Zn k_3 -weighted EXAFS for ZnPor (a), ZnPor-RuCuDAC (b), ZnPor-Ru₂DAC (c) and ZnPor-Cu₂DAC (d).

Supplementary Figure 12. Wavelet transforms of Ru k_3 -weighted EXAFS for RuN₃ (a) and Cu k_3 -weighted EXAFS for CuN₃ (b).

8. *The mark of Supplementary Figure 10a is incorrect.*

Re: Thanks a lot for this useful suggestion!

Accordingly, former Supplementary Figure 10a has been corrected, which is labelled as Supplementary Figure 11a in the revised manuscript.

9. *According to the structure (Supplementary Figure 4) and the unit cell parameters given by the author, the distance between the Ru and Cu/Ru atoms in the upper and lower layers also seems to be very close (3.81 or 3.81/2), and all Ru and Cu atoms are in the state of overlapping. Why is this signal not observed in EXAFS? Does this also mean that the author's analysis of the COF skeleton structure needs further verification?*

Re: We thank for bringing this important idea to our attention

As shown in Figure R1, the ionic radius of Zn^{2+} , Ru^{2+} and Cu^{2+} is 0.74, 0.81 and 0.72 Å, respectively, and the optimized unit cell parameters show ZnPor-RuCuDAC COF has a layer distance of 3.81 Å, suggesting that Zn^{2+} , Ru^{2+} and Cu^{2+} in the adjacent layers are separated theoretically for enough interlayer distance (ca. 2.2–2.4 Å). On the other hand, the strong in-layer conjugated system caused by the porphyrin and pyridine groups can further hold back the interaction of Ru and Cu/Ru atoms in the upper and lower layers.

In fact, many references of porphyrin-based polymer or COF for photo(electron)catalysts have reported that there is not atom overlapping in the interlayer even though these COFs take a layer distance ca. 4 Å (Nat. Commun. 2021, 12, 1354; Adv. Funct. Mater. 2021, 31, 2107290; J. Am. Chem. Soc. 2021, 143, 18052-18060; Angew. Chem. Int. Ed. 2020, 59, 3624-3629). For example, Chen, R. F. et al. (refer to: Nat. Commun. 2021, 12, 1354) reported the ZnPor-DETH-COF with a layer distance of 3.8894 Å; Wang, T. X. et al (refer to: Adv. Funct. Mater. 2021, 31, 2107290) synthesized a CuPor-based COF with 4.0 Å layer distance; Yue y. et all (refer to: J. Am. Chem. Soc. 2021, 143, 18052-18060) reported CuPcF8-CoPc-COF with 3.4322 Å layer distance. Taking the above consideration, we would like to think Ru and/or Cu atoms of those COF photocatalysts possibly exist the interaction at the same layer better rather than the interlayers if this reviewer agrees. Also, based on this interesting comment, we will further investigate the atom interlayer interaction of atom-site catalysts in the future research by chemical small molecular modification.

H 32																	B 82	C 77	N 70	O 66	F 64	
Li 123 M ⁺ 60	Be 89 M ²⁺ 31																	M ²⁺ 20	M ⁴⁺ 16	M ³⁺ 171 M ²⁺ 11	M ²⁺ 140 M ⁶⁺ 9	x ⁻ 136
Na 154 M ⁻ 95	Mg 136 M ²⁺ 65																	Al 118 M ³⁺ 50	Si 117 M ⁴⁺ 42	P 110 M ²⁺ 212 M ³⁺ 34	S 104 M ²⁺ 184 M ⁶⁺ 29	Cl 99 x ⁻ 181
K 203 M ⁻ 133	Ca 174 M ²⁺ 99	Sc 144 M ²⁺ 81	Ti 132 M ²⁺ 90 M ²⁺ 76 M ⁴⁺ 68	V 122 M ²⁺ 88 M ²⁺ 74	Cr 118 M ²⁺ 84 M ²⁺ 69	Mn 117 M ²⁺ 80 M ²⁺ 66	Fe 117 M ²⁺ 76 M ²⁺ 64	Co 116 M ²⁺ 74 M ²⁺ 63	Ni 115 M ²⁺ 72 M ⁴⁺ 62	Cu 117 M ⁻ 96 M ²⁺ 72	Zn 125 M ²⁺ 74	Ga 126 M ⁻ 113 M ²⁺ 62	Ge 122 M ⁴⁺ 53 M ²⁺ 73	As 121 M ³⁺ 222 M ²⁺ 69 M ⁴⁺ 47	Se 117 M ²⁺ 198 M ⁶⁺ 42	Br 114 x ⁻ 195						
Rb 216 M ⁻ 148	Sr 191 M ²⁺ 113	Y 162	Zr 145 M ⁴⁺ 80	Nb 134 M ²⁺ 70	Mo 130 M ⁶⁺ 62	Tc 127	Ru 125 M ²⁺ 81	Rh 125 M ²⁺ 80	Pd 128 M ²⁺ 85	Ag 134 M ⁻ 126 M ²⁺ 89	Cd 148 M ²⁺ 97	In 144 M ⁻ 132 M ²⁺ 81	Sn 140 M ⁴⁺ 71 M ²⁺ 93	Sb 141 M ²⁺ 245 M ²⁺ 92 M ⁴⁺ 62	Te 137 M ²⁺ 221 M ⁶⁺ 56	I 133 x ⁻ 216						
Cs 235 M ⁻ 169	Ba 198 M ²⁺ 135	La-Lu	Hf 144 M ⁴⁺ 79	Ta 134 M ²⁺ 69	W 130 M ⁶⁺ 62	Re 128	Os 126 M ²⁺ 88	Ir 127 M ²⁺ 92	Pt 130 M ²⁺ 124	Au 134 M ⁻ 137 M ²⁺ 85	Hg 144 M ²⁺ 110	Tl 148 M ⁻ 140 M ²⁺ 95	Pb 147 M ⁴⁺ 84 M ²⁺ 120	Bi 146 M ³⁺ 108 M ²⁺ 74	Po 146	At 145						
La 187.7 M ²⁺ 106.1	Ce 182.4 M ²⁺ 103.4 M ³⁺ 92	Pr 182.8 M ²⁺ 101.3 M ⁴⁺ 90	Nd 182.1 M ²⁺ 99.5	Pm 181.0 M ²⁺ 97.9	Sm 180.2 M ²⁺ 111 M ³⁺ 96.4	Eu 204.2 M ²⁺ 109 M ³⁺ 95.0	Gd 180.2 M ²⁺ 93.8	Tb 178.2 M ²⁺ 92.3 M ⁴⁺ 84	Dy 177.3 M ²⁺ 90.8	Ho 176.6 M ²⁺ 89.4	Er 175.7 M ²⁺ 88.1	Tm 74.6 M ²⁺ 94 M ³⁺ 86.9	Yb 194.0 M ²⁺ 93 M ³⁺ 85.8	Lu 173.4 M ²⁺ 84.8								

Figure R1. Atom and ionic radius of element (for review only).

10. How does the author determine that only Ru-Cu pairing exists in ZnPor-RuCuDAC, without Ru-Ru pairing and Cu-Cu pairing? The EXAFS fitting results show that the length of the Ru-Cu bond in ZnPor-RuCuDAC is 2.70, and the length of the Ru-Ru bond in ZnPor-Ru₂DAC is 2.68. This seems indistinguishable by fitting.

Re: Thanks for the useful comment.

The Ru–Cu sites of ZnPor-RuCuDAC are not just confirmed from the bond length Ru–Cu (2.70 Å). As seen in the atomic-resolution ADF-STEM images of ZnPor-RuCuDAC, the 3D electron diffraction tomography of DPC images (Fig. 1h–j) show that Ru and Cu elements on the ZnPor-RuCuDAC backbones are the atomic cross-distribution with uniform and continuous profiles, which the Ru and Cu atoms do not show a phase difference on the spatial location. Therefore, taking the results of XAS and XPS together, it may be better to consider that ZnPor-RuCuDAC mainly exist Ru-Cu site even though few homonuclear appear possibly.

Accordingly, the corresponding description (page 9, 2nd paragraph) is revised: “However, the Ru centers of ZnPor-RuCuDAC (Fig. 2g,h) and ZnPor-Ru₂DAC (Fig. 2i,j) show maximum intensity for the Ru–N and Ru–metal paths, compared to RuN₃ (Supplementary Fig. 12a). Further, both ZnPor-RuCuDAC (Fig. 2k,l) and ZnPor-Cu₂DAC (Fig. 2m,n) exhibit Cu–N and Cu–metal shells, unlike CuN₃ reference (Supplementary Fig. 12b). Also, as compared to ZnPor and RuN₃/CuN₃ monomers, all COFs present Zn(II) 2p_{3/2}/2p_{1/2} binding energy (BE)

peaks ascribable to Zn-N bond (Supplementary Fig. 13), but emerge new BE peaks of Ru-Cu (ZnPor-RuCuDAC), Ru-Ru (ZnPor-Ru₂DAC) and Cu-Cu (ZnPor-Cu₂DAC) diatomic pairs besides Ru-N and/or Cu-N bond.¹⁵ These results collectively indicate that compared to the single-atom Zn distribution, ZnPor-RuCuDAC mainly exist Ru-Cu diatomic pairs via diatomic coordination assemblies of metal-N₃ connected by Ru-Cu bonding bridges even though few homonuclear appear possibly, while ZnPor-Ru₂DAC and ZnPor-Cu₂DAC are anchored the Ru-Ru and Cu-Cu sites.”

11. In the calculation of band structures, the values (1.05, 0.95, and 0.93 eV) obtained from the SRPES valence band spectra (Supplementary Figures 13a-c) should be the energy difference between the VBM and Fermi level (EVBM-E_f), not the EVBM. The band structure analysis needs to be corrected.

Re: Thanks a lot for this useful comment!

After carefully checking those valence band spectra derived from synchrotron-radiation photoemission spectroscopy (SRPES) measurement and consulting the SRPES technician, we found that those VB values are correct, but we are very sorry for our carelessness on the descriptions of SRPES valence band spectra. Factually, although the SRPES valence band spectra is the energy difference between the VBM and Fermi level, the valence-band spectra are measured using synchrotron-radiation light as the excitation source with photoenergy of 40.00 eV and referenced to the Fermi level ($E_f = 0$) determined from Au (please refer to: J. Am. Chem. Soc. **2016**, 138, 8928–8935; J. Am. Chem. Soc. 2012, 134, 7600–7603; Carbon 2020, 157, 340–349).

To make this issue clearer, the corresponding part in the supporting information was revised as: “Under an ultrahigh vacuum (UHV) chamber, ultraviolet photoelectron spectroscopy (UPS) was obtained by synchrotron radiation photoemission spectroscopy (SRPES) measurements. The valence-band spectra were measured using synchrotron-radiation light as the excitation source with photon energy of 40.00 eV and referenced to the Fermi level ($E_f = 0$) determined from Au. A sample bias of –5 V was applied in order to observe the secondary electron cutoff.”

12. This new band may be attributed to the deformation vibration of Ru-Cu sites, containing the absorbed C-C coupling intermediates during CO₂RR. Whether there is a certain proof basis for the description of the C-C coupling intermediates, it is suggested to consider citing literature.

Re: Thanks a lot for this useful comment!

Some related literatures (Angew. Chem. Int. Ed. 2022, 61, e202208904; Nature 2021, 600, 81–85; ACS Nano 2020, 14, 11363-11372) are cited in the revision and listed in Reference as Ref. 30,46,47 according to their corresponding research contents, which can help to understand the new Raman vibration band (~202–216 cm⁻¹) at Ru-Cu sites during 0–90 min of light irradiation in reference to the ground-state spectrum besides further blue-shifting peaks. Accordingly, the original references were also changed in both manuscript and reference list. For the details, please refer to document named as “Highlighted revision of NCOMMS-23-00556”.

13. It is recommended to add literature support for signal attribution in DRIFTS.

Re: Thanks a lot for this helpful suggestion.

We have replaced some literatures to support the DRIFTS result (cited as: 5, 9, 11, 15, 17,29).

5. Hu, Y. G. et al. Tracking mechanistic pathway of photocatalytic CO₂ reaction at Ni sites using operando, time-resolved spectroscopy. *J. Am. Chem. Soc.* **142**, 5618–5626 (2020).
9. Ran, L. et al. Engineering single-atom active sites on covalent organic frameworks for boosting CO₂ photoreduction. *J. Am. Chem. Soc.* **144**, 17097–17109 (2022).
11. Zhang, Y. Z. et al. Atomic-level reactive sites for semiconductor-based photocatalytic CO₂ reduction. *Adv. Energy Mater.* **10**, 1903879 (2020).
- 15 Wang, J. M. et al. Highly durable and fully dispersed cobalt diatomic site catalysts for CO₂ photoreduction to CH₄. *Angew. Chem. Int. Ed.* **61**, e202113044 (2022).
17. Liu, Q. et al. Regulating the *OCCHO intermediate pathway towards highly selective photocatalytic CO₂ reduction to CH₃CHO over locally crystallized carbon nitride. *Energy Environ. Sci.* **15**, 225–233 (2022).
29. Wang, W. et al. Photocatalytic C-C coupling from carbon dioxide reduction on copper oxide with mixed-valence Copper(I)/Copper(II). *J. Am. Chem. Soc.* **143**, 2984–2993 (2021).

14. *When the interaction between Ru-Cu sites during the photocatalytic process is calculated, why not consider the interaction of active sites between adjacent layers?*

Re: Thanks for this constructive suggestion.

As seen in the revision of the second reviewer 9th comment, after considering the metal ionic size, the Zn²⁺, Ru²⁺ and Cu²⁺ in the adjacent layers are possibly separated for ca. 2.2–2.4 Å interlayer distance. Moreover, the strong Coulomb repulsion among the interlayer/interspace, which is derived from the strong in-layer conjugation of porphyrin and pyridine groups, can also lead to a weak metal-active-site interaction between the upper and lower layers. On the DFT calculation, the interaction between diatomic site and intermediates are confined in the in-layer atom sites rather than adjacent layers (please refer to: *Am. Chem. Soc.* **2022**, *144*, 7822-7833; *J. Am. Chem. Soc.* **2021**, *143*, 18052-18060; *Adv. Energy Mater.* **2021**, *11*, 2003575). In fact, the interaction of active sites between adjacent layers maybe just result to a difference photogenerated electron distribution, and this work focus on the same layer atom site more. When understanding the interaction mechanism between active site and intermediates, the effect of sites on the adjacent sites are negligible for simplified calculation process (referring to: *Angew. Chem. Int. Ed.* **2022**, *61*, e202208904; *J. Am. Chem. Soc.* **2018**, *140*, 6474-6482; *J. Am. Chem. Soc.* **2020**, *142*, 16723-16731).

15. *It seems that the author synthesized a COF material after reading the full text. Why is it called an intraskeletal photogate molecular device (PMD) catalyst?*

Re: Many thanks for this professional comment!

CO₂ photoreduction typically involves the process of photo-absorption, carrier separation, and CO₂ reduction. The former two processes are closely related to conversion efficiency, while the third mainly determines the final selectivity. Therefore, tremendous descriptors have been proposed for developing various photosystems to modulate the activity of catalytic centers, light-harvesting ability of chromophores, and efficiency of charge transfer between photosensitizers and catalytic centers. Among which, because integrating the chromophoric photosensitizers, catalytic sites and electron-transfer mediates to form a full-component molecular photosystem, intraskeletal photogate molecular device (PMD)-derived catalysts are recognized as promising candidates. However, most of these PMD-like catalysts reported for

light-driven CO₂RR merely worked in homogeneous systems, resulting in various conundrums for large-scale applications, including solvent-dependence solubility, unsatisfactory quantum efficiency, and vulnerability under harsh photocatalytic conditions.

Taking for a new class of crystalline porous materials, covalent organic frameworks (COFs), bridged by molecular organic units, have been developed as heterogeneous photocatalysts for CO₂RR with merits of structural regularity, synthetic tunability, and high stability. However, the photoactive and catalytic centers in COFs are isolated in space, and long-distance electron migration is challenging owing to a weak electronic coupling.

To overcome these above problems, an alternative strategy is to integrate the close-lying PMDs into COF matrices that allow intraskeletal electron transfers between the directly bonded photosensitizers and catalytic centers at the COF compartments, facilitating an efficient energy transfer and electron diffusion during the CO₂RR photocatalysis. Nevertheless, this type of COF photocatalyst has been rarely explored to date.

Inspired by the above considerations, we synthesized heteronuclear dual-atom-site COFs by self-assembling Zn-porphyrin and Ru/Cu-pincer complexes for photocatalytic CO₂RR, where every Zn-porphyrin photoresponsive center is surrounded by eight Ru/Cu catalytic units, forming a highly-ordered square structure that facilitates PMD-type energy transfer and electron collection pathways.

To better understand this work, the relative part of introduction (page 3, 1st paragraph) is revised as: “Comparatively, because of the distinct electronegativity and gradient orbital coupling at the heteroatoms, the intermediates on heteronuclear DACs possibly exhibit asymmetric charge distributions, which can suppress the repulsive molecular interactions of the formed intermediates and augment the collision probability of adjacent intermediates to couple and form C₂₊ products. **A promising strategy is to merge the close-lying PMDs into covalent organic frameworks (COFs) with diatomic sites, facilitating an efficient energy transfer between the bonded photosensitizers and dual-atom centers at the COF compartments.** Therefore, an in-depth insight into the selectivity mechanism of heteronuclear and homonuclear diatomic sites is necessary to reveal the unique advantages of DACs, aiding in the rational design of PMD-derived photocatalysts with high catalytic performances and multifunctionalities via structural and spatial isomerization.”

Reviewer #3 (Remarks to the Author): This manuscript reported photocatalytic CO₂ reduction on Ru-Cu couple site on COF and the results are highly interesting because acetic acid was selectively formed in spite of difficulty of C-C bond formation by photocatalyst. The content is well organized however the following points are still not clear and revision is required on the following points.

1. Stability is still wondering because COF is consisted of Zn-porphyrin and pyridine with dehydration reaction. Since photocatalytic reaction is performed under coexistence of triethanolamines which may be alkaline condition, hydration of this C=N bond may expect to be dissociated during reaction. Although authors show XRD after reaction, broadening of the peaks was recognized and also please add intensity for Y axis in supplementary Fig.20 (b).

Re: Thanks a lot for this useful comment!

The PXRD Y axis of ZnPor-RuCuDAC after 40 h photoreaction is modified in the revision at Supplementary Fig. 23b. To confirm the chemical-structure stability of these COFs in acid/alkaline solution, we have conducted a series of characterizations on the revision. Firstly, the chemical-stable crystallinity of original ZnPor-RuCuDAC is explored by dispersing it into different solvents (*n*-hexane, 0.5 M HCl, 6.0 M NaOH). Notably, when comparing with the raw sample, the immersed ZnPor-RuCuDAC exhibit intense PXRD patterns at 3.06°, 4.42°, and 9.57° (Supplementary Figure 8a), without significant change in the peak position and intensity. In addition, the detection results of ICP-OES confirmed that there was no Zn/Ru/Cu element signal in the treatment supernatant, which was centrifugally separated from the ZnPor-RuCuDAC suspension after treated with these different solutions for 40 h. These results indicated that the highly chemical-stable crystallinity was retained under acid/alkaline conditions.

Secondly, to evaluate the stability of chemical structure after four consecutive runs for a total 40 h irradiation, XPS spectra of these recovered diatomic COFs are added as Supplementary Fig. 24 in supporting information. The survey XPS spectra of the recovered COFs after photocatalytic reaction indicates that Zn, Ru, Cu, C and N still coexist in the COFs' skeleton. The high-resolution recovered XPS spectra has the binding energy peaks of Zn^{II} 2p_{3/2}/2p_{1/2}, Ru^{II} 3p_{3/2}/3p_{1/2} and Cu^{II} 2p_{3/2}/2p_{1/2}, which are very similar to that of the original one. These analyses indicate that the element composition and valence states of diatomic composite

are unchanged during the long-term photoreaction process, and thus has very high structural stability.

Third, the DRS and PXRD were also tested toward the recovered ZnPor-RuCuDAC photocatalyst. As shown in the Supplementary Fig. 23 at supporting information, the DRS spectra of revived ZnPor-RuCuDAC show strong absorption intensity in the range of ~300–817 nm, which is very similar to the original ZnCu-COF absorption. PXRD of the recovered ZnPor-RuCuDAC present four main diffraction peak at 3.06° , 4.42° , and 9.57° , corresponding to (202), (200) and (211) crystal facets, respectively, which match well with the protogenous sample. Based on the above considerations, these diatomic COF composite have stable chemical structure during the long-term photoreaction process under alkaline condition.

According to this comment, the related part (page 6, 1st paragraph) in revised manuscript was revised as: “The porosity of these COFs was evaluated from the nitrogen sorption isotherms (Supplementary Fig. 5–7). The Brunauer–Emmett–Teller (BET) surface areas were estimated as 902.3 (ZnPor-RuCuDAC COF), 867.3 (ZnPor-Ru₂DAC COF), and 826.3 (ZnPor-Cu₂DAC COF) m² g⁻¹, with pore volumes of 0.36, 0.31, and 0.25 cm³ g⁻¹, respectively. The measured pore sizes were 1.66 (ZnPor-RuCuDAC COF), 1.68 (ZnPor-Ru₂DAC COF), and 1.65 (ZnPor-Cu₂DAC COF) nm, further confirming the AB-staggered stacking geometry of these COFs due to the complete agreement between experimental and theoretical pore sizes. **As contrasted to the original PXRD pattern, ZnPor-RuCuDAC (Supplementary Fig. 8a) do not vary in the peak position and intensity under different solvents obviously, indicating highly chemical-stable crystallinity under acid/alkaline conditions. From thermogravimetric analysis (Supplementary Fig. 8b–d), all three COFs are thermally stable up to 368 °C.**”

The related part (page 11, 2nd paragraph) in revised manuscript was revised as: “In addition to the high acetate selectivity of ZnPor-RuCuDAC, a stable photoactivity was also retained (Fig. 3f and Supplementary Fig. 20–22), and no obvious decay was observed during four consecutive runs. **Moreover, the diatomic catalyst and its recovered composite present quite similar DRS (Supplementary Fig. 23a), PXRD (Supplementary Fig. 23b) and XPS spectra (Supplementary Fig. 24), suggesting that crystallinity, element composition and oxidation state are still maintained after 40 h reaction....**”

Supplementary Figure 8. (a) PXRD pattern of ZnPor-RuCuDAC before and after treated in different solvents for 40 h. In a typical experiment, 10 mg of ZnCu-COF was immersed in the solvents (n-hexane, 0.5 M HCl, 6.0 M NaOH) at room temperature.

Supplementary Figure 24. Survey (a), high-resolution Zn2p (b), Ru3p (c) and Cu2p (d) XPS spectra of ZnPor-RuCuDAC, ZnPor-Ru₂DAC, ZnPor-Cu₂DAC after 40 h photoreaction, respectively.

2. Cl impurity may be contained, and I wonder Cl content in catalyst may be negative impact or prevent further oxidation of acetic acid.

Re: Many thanks for this comment!

We agree with the opinion that Cl impurity may impact the photoreaction process, and sorry for our careless descriptions on this issue. The RuN₃ and CuN₃ pincer complex were synthesized via [RuCl₂(p-cymene)]₂ and [CuCl₂(p-cymene)]₂ precursors, respectively, and thus the obtained products possibly exist impurity Cl. However, the ultimate COFs' product was collected by centrifugation, and exhaustively washed by Soxhlet extractions with methanol and tetrahydrofuran for 72 h. Finally, the brown diatomic COFs was dried under vacuum oven at 80 °C. This process can basically remove the Cl impurity. On the other hand, the results of elemental analysis indicate that no Cl impurity can be detected on the ZnPor-RuCuDAC, ZnPor-Ru₂DAC and ZnPor-Cu₂DAC COF. Also, Ru K-edge spectra of RuN₃ precursor show coordination peaks of Ru–Cl (1.99 Å), and the *R*-space Fourier transformed Cu K-edge spectrum of CuN₃ precursor exhibits peaks at 2.07 Å corresponding to the Cu–Cl shell scatterers. However, comparatively, the Ru–Cl and/or Cu–Cl signals disappear in the K-edge spectra of the ZnPor-RuCuDAC, ZnPor-Ru₂DAC, and ZnPor-Cu₂DAC backbones, indicating the Cl⁻ disassociation from the Ru and/or Cu sites during the condensation of the ZnPor, RuN₃, and CuN₃ monomers.

To avoid the misunderstanding toward Cl impurity, the relative part (page 9, 1st paragraph) in revised manuscript is revised as: “However, the Ru–Cl and/or Cu–Cl signals disappear in the spectra of the ZnPor-RuCuDAC, ZnPor-Ru₂DAC, and ZnPor-Cu₂DAC backbones, because of the Cl⁻ disassociation from the Ru and/or Cu sites during the condensation of the ZnPor, RuN₃, and CuN₃ monomers, indicating no Cl impurity on the COFs in consistent with element analysis.”

3. P.12, Authors claim Ru-Cu stretching and bending mode were observed at 226 and 252 cm⁻¹, respectively in Raman spectroscopy without any proof and so I suggest authors to simulate Raman spectrum based on the model proposed in Supplementary Scheme 1 for demonstrating. Raman spectroscopic data after measurement is also required to confirm Ru-Cu DAC sustained.

Re: Thanks a lot for this professional suggestion.

The simulated Raman spectrum of ZnPor-RuCuDAC has been added to the revision (Supplementary Fig. 28). Based on the vibration-frequency model analysis of molecular structure, the peak at ~226 and 252 cm⁻¹ can be ascribable to the Ru–Cu stretching and bending

vibration, respectively, which are consistent with the experimental values. On the other hand, as seen in Quasi *in-situ* Raman spectra of ZnPor-RuCuDAC after measurement (Fig. 4b, under 0–30 min light-off condition), the Ru–Cu deformation vibration band disappears, and the Ru–Cu stretching and Ru–Cu bending vibration peaks return to their original states within 30 min light-off, indicating that Ru–Cu diatomic site show favorable sustainability during CO₂ photoreduction.

According to comment, the Raman analyses part (page 13) is revised as: “As shown in Fig. 4a, the peak positions of both the Ru–Cu stretching ($\sim 226\text{ cm}^{-1}$) and bending ($\sim 252\text{ cm}^{-1}$) vibrations are consistent with the simulated one (Supplementary Fig. 28), which gradually shift to higher wavenumbers during CO₂ adsorption and show no change at 60 min for the saturated CO₂ adsorption. The observed 3-nm blue-shifting indicates that the orbital coupling between M-d (M represents Ru or Cu metal atom) and C-2p orbitals promote electron cloud migration from the metal sites to the CO₂ orbital, leading to CO₂ molecular activation. Under illumination, the enriched photoelectron cloud of the Ru–Cu sites further moves to the absorbed CO₂ molecular orbitals for the photoreduction of CO₂ and reaction intermediates, resulting in Ru–Cu bonds with changeable strengths that again promote Ru–Cu vibration peaks shifting to higher frequencies (Fig. 4b).⁴⁶ Furthermore, as seen in the variation of Raman peak position obtained in different measurement (Fig. 4c, blue rectangle of the light-on and light-off spectra), a Raman vibration band ($\sim 202\text{--}216\text{ cm}^{-1}$) at Ru–Cu sites emerges during 0–90 min of light irradiation in reference to the ground-state spectrum besides further blue-shifting peaks. This new band may be attributed to the deformation vibration of Ru–Cu sites, containing the absorbed C–C coupling intermediates during CO₂RR.^{30,46,47} After measurement (Fig. 4b,c), the deformation vibration band disappears, and the Ru–Cu stretching and Ru–Cu bending vibration peaks return to their original states within 30 min light-off, suggesting favorable reversibility of Ru–Cu diatomic site.”

Supplementary Figure 28. (a) Simulated Raman spectra of ZnPor-RuCuDAC and (b) the zoomed Raman peaks in the region of 0–500 cm^{-1} of Supplementary Figure 26(a). According to vibration-frequency analysis of molecular structure, the peak at ~ 226 and 252 cm^{-1} can be ascribable to the Ru–Cu stretching and bending vibration, respectively.

4. Fig.4, Temperature is missing and if this is room temperature, adsorption of CO is quite weak and so I wonder such weak adsorption species can contribute to reaction.

Re: Thanks the referee for professional comment!

The *quasi in-situ* Raman spectra of ZnPor-RuCuDAC, and *In-situ* DRIFTS profiles of CO desorption and CO_2RR of these diatomic COFs at different times are conducted on the same $53 \text{ }^\circ\text{C}$ temperature as photocatalytic CO_2 reduction experiment. Factually, even though the adsorption of CO is quite weak on the active site, the spectra of *quasi in-situ* Raman and DRIFTS profiles toward photo(electron)catalytic catalysts can be obtained under room temperature by background-correction method (please refer to: ACS Nano 2020, 14, 11363–11372; Nature 2021, 600, 81–85; J. Am. Chem. Soc. 2020, 142, 5618–5626; J. Am. Chem. Soc. 2022, 144, 17097–17109).

Accordingly, the related part (page 12) of *quasi in-situ* Raman analysis is revised as: “*Quasi in-situ* Raman spectroscopic analyses were carried out to identify possible reactive sites on diatomic COFs under the same $53 \text{ }^\circ\text{C}$ temperature as photocatalytic CO_2 reduction experiment.”

The related part (page 14) of DRIFTS spectra is revised as: “Photocatalytic CO_2RR occurs in multiple steps involving many intermediates, among which $^*\text{CO}$ is an important one. Notably, the $^*\text{CO}$ desorption results in the CO formation; protonated $^*\text{CO}$ evolves in CH_4 ,

whereas the dimerization of two *CO intermediates lead to the formation of C_2 products.^{15,17,29} Therefore, to unveil the intrinsic adsorption behavior of the *CO intermediates, diffuse reflectance infrared Fourier-transform spectroscopy (DRIFTS) was conducted on the heteronuclear/homonuclear diatomic COFs **under 53 °C.**”

5. Fig.5 and discussion on reaction route; It seems CO coupling is the rate determining step and CO hydrogenation is not large activation energy required. If so, why CH_4 or coupling of CH_3 species of C_2H_6 are not formed. From energy level calculation, CH_3COOH energy level is almost the same with that of CO_2 and this is highly strange and not reliable.

Also, adsorption model of CO on Ru and Cu DAC site is same direction and what data does support of this adsorption model and why such orientation is occurred because opposite site seems to be more reasonable form energy minimization.

Re: Thank you so much for valuable comments!

We are very sorry for our ambiguous descriptions on the possible formation of CH_4 and C_2H_6 on these diatomic catalysts. To clearly discuss the possibility, the CO_2 reduction pathway to CH_4 is listed as following, where the excited states of (*CO_2) are considered as the zero initial states at the electron-proton transfer process:

First, as considering the reduction pathways of C_1 or C_2 formation, the *CO is a crucial

intermediate that is derived from initial CO₂ activation. Notably, the *CO desorption results in the CO formation; protonated *CO evolves in CH₄, whereas the dimerization of two *CO intermediates lead to the formation of C₂ products. As seen in Fig. 5a and Supplementary Fig. 39, on the heteronuclear Ru–Cu site of ZnPor-RuCuDAC, the two *CO coupling energy barrier is 0.51 eV to form *OC–CO, the *CO desorption energy is 1.02 eV to form CO gas, and the ΔG of *CO protonation to (*HCO+*HCO) or (*HOC+*HOC) that can be protonated to CH₄ and C₂H₆ is 1.56 eV or 2.28 eV, respectively. These ΔG values indicate that the *OC–CO intermediates, formed on heteronuclear Ru–Cu sites, are more thermodynamically favored than the *CO desorption and *CO hydrogenation to *CHO or *COH. Because of the higher protonation energy barrier than two *CO coupling, CO, HCOOH, CH₄ or C₂H₆ would be difficult to produce in the ZnPor-RuCuDAC, which the acetate is confirmed the main product and few CO and formate are detected. Factually, because of multi-step protonation and severe dehydration (dehydroxylation), CH₄ and C₂H₆ are difficult to form in photocatalyst system, especially in liquid-phase catalytic system (Angew. Chem. Int. Ed. 2021, 60, 24849-24853; J. Am. Chem. Soc. **2018**, 140, 6474-6482; Nature 2020, 586, 549-554), and so the main product have been reported as CO, HCOOH, CH₃OH, CH₃CH₂OH and CH₃COOH.

Similarly, on the homonuclear Ru–Ru site of ZnPor-RuRuDAC (Supplementary Fig. 40), the *CO desorption energy is 0.58 eV to form CO gas, the two *CO coupling energy barrier is 1.52 eV to form *OC–CO, and the ΔG of *CO protonation to (*HCO+*HCO) or (*HOC+*HOC) is 1.73 eV or 2.33 eV, respectively. These ΔG values suggest that the *CO desorption on homonuclear Ru–Ru sites, are more thermodynamically favored than the *CO coupling and *CO hydrogenation to *CHO or *COH because of the high energy barrier. Moreover, on the homonuclear Cu–Cu site of ZnPor-CuCuDAC (Supplementary Fig. 41), the *CO desorption energy to form CO gas is exothermic (-0.46 eV), the two *CO coupling energy barrier is 0.58 eV to form *OC–CO, and the ΔG of *CO protonation to (*HCO+*HCO) or (*HOC+*HOC) is 1.36 eV or 2.17 eV, respectively. These ΔG values reveal that the *CO desorption on homonuclear Cu–Cu sites, are more thermodynamically favored than the *CO coupling and *CO hydrogenation.

According to the above ΔG comparison of different reaction pathways for adsorbed *CO + *CO on these diatomic COFs, the two *CO coupling to *OC–CO intermediates on

heteronuclear Ru–Cu sites, are more thermodynamically favored than the *CO desorption and *CO hydrogenation to *CHO and then to CH_4 . In contrast, the Gibbs free energy of *CO desorption to CO gas on ZnPor-Ru₂DAC is smaller than that for *CO protonation, and ZnPor-Cu₂DAC promotes an exothermic CO formation rather than an endothermic C–C coupling and *CO hydrogenation, these ΔG results are consistent with the photoactivity.

Second, Fig. 5a show the Gibbs free energy diagrams of the deduced reaction pathways over these diatomic COFs, where the energy level of every step just refers to its former intermediate (the detailed information refers to the first reduction pathways of CO₂ in Supporting Information). The free energy changes (ΔG) relative to an former state can be obtained by the energy difference between products and reactants at every step. Because of the energy of ($e^- + H^+$), H₂O desorption and intermediate isomerization are contained from step-I to step-XII under DFT calculation, it is meaningless to compare the CH₃COOH energy level with the initial state of CO₂. For example, CH₃COOH energy level is -0.31 eV, obtained from the step-XII ΔG ($^*HOOC-CH_3 \rightarrow ^* + HOOC-CH_3$), which is meaningful just relative to its $^*HOOC-CH_3$ (-0.79 eV) precursor, indicating that the $^*HOOC-CH_3$ intermediate desorption is endothermic to form ($^* + HOOC-CH_3$) on the diatomic site. Relative to the CO₂ initial state, CH₃COOH energy level is -0.31 (ZnPor-RuCuDAC), 0.23 (ZnPor-Ru₂DAC) and 0.69 (ZnPor-Cu₂DAC) eV, respectively. This just suggest that from CO₂ to CH₃COOH, the overall catalytic system is exothermic (-0.31 eV) on heteronuclear Ru–Cu site of ZnPor-RuCuDAC, while the catalytic system of ZnPor-Ru₂DAC and ZnPor-RuCuDAC is endothermic.

Third, to analyze the low C–C coupling reaction barrier of the Ru–Cu sites of ZnPor-RuCuDAC, we calculated the charge density difference on these diatomic COFs after the formation of two *CO intermediates. When conducting on the first-principles calculations, adsorption model is built on the base of unit cell parameters, and its structure will be optimized with the absorbed intermediate. After geometry optimization, the adsorption model of CO these DAC site is obtained. From the structural parameters (Supplementary Fig. 44) of two *CO on Ru–Cu site of ZnPor-RuCuDAC in the revision, the two *CO intermediates nearly appear parallel to each other on the heteronuclear Ru–Cu atoms with a side-to-side configuration and a negligible 4.7° twist along the dual-atom direction (not the same direction along dual-atom axis). This twist is different from the broad dihedral angle distortion of the Ru–Ru (19.5°) and

Cu–Cu (27.1°) sites (Supplementary Fig. 45,46). The small dihedral angle distortion is ascribable to a weak molecular repulsion between the two *CO intermediates on the Ru–Cu site, resulting that the adsorption-model system of two *CO on Ru–Cu site has the lowest energy.

According to this comment, the corresponding description (page 17, 1st paragraph) in revised manuscript is revised: “...From CO₂ to CH₃COOH, the overall catalytic system is exothermic (-0.31 eV) on heteronuclear Ru–Cu site, while the catalytic system of ZnPor-Ru₂DAC (0.23 eV) and ZnPor-RuCuDAC (0.69 eV) is endothermic. Therefore, the *CO intermediate on ZnPor-Cu₂DAC is inclined to desorb as CO rather than couple to the *OC–CO as it does on the Ru–Cu site. This inferior C–C coupling accounts for the very low yield of acetate on homonuclear diatomic COFs.”

The corresponding description for Supplementary Fig. 39 is revised: “...The two *CO coupling energy barrier is 0.51 eV to form *OC–CO, the *CO desorption energy is 1.02 eV to form CO gas, and the ΔG of *CO protonation to (*HCO+*HCO) or (*HOC+*HOC) that can be protonated to CH₄, C₂H₆, etc is 1.56 eV or 2.28 eV, respectively. These ΔG values indicate that the *OC–CO intermediates, formed on heteronuclear Ru–Cu sites, are more thermodynamically favored than the *CO desorption and *CO hydrogenation to *CHO or *COH.”

The description for Supplementary Fig. 40 is revised: “...The *CO desorption energy is 0.58 eV to form CO gas, the two *CO coupling energy barrier is 1.52 eV to form *OC–CO, and the ΔG of *CO protonation to (*HCO+*HCO) or (*HOC+*HOC) is 1.73 eV or 2.33 eV, respectively. These ΔG values suggest that the *CO desorption on homonuclear Ru–Ru sites, are more thermodynamically favored than the *CO coupling and *CO hydrogenation to *CHO or *COH.”

The description for Supplementary Fig. 41 is revised: “...The *CO desorption energy to CO gas is exothermic (-0.46 eV), the two *CO coupling energy barrier is 0.58 eV to form *OC–CO, and the ΔG of *CO protonation to (*HCO+*HCO) or (*HOC+*HOC) is 1.36 eV or 2.17 eV, respectively. These ΔG values reveal that the *CO desorption on homonuclear Cu–Cu sites, are more thermodynamically favored than the *CO coupling and *CO hydrogenation.”

6. No detail information of oxidation species and I think triethanolamine may be used for sacrificial agent, however, if so, ethanol may form and further oxidized to CH₃COOH. However,

from tracer experiment using ^{13}C , the authors claimed only ^{13}C contained CH_3COOH was formed and if so, what is oxidation species and how to consume hole?

Re: Thanks for this professional comment!

After carefully reading some references (Chem. Eur. J. 2011, 17, 12891; Inorg. Chem. 2008, 47, 0378–10388; C. R. Chimie 2017, 20, 283-295; J. Am. Chem. Soc. 1980, 102, 5627–5631; Acc. Chem. Res. 1980, 13, 83–90), the possible mechanism of triethanolamine (TEOA) degradation pathway as sacrificial reagent in present photocatalytic system are discussed in the revision (Supplementary Fig. 17). Once contacting with the excited ZnPor photosensitive centers, a positively charged aminyl radical is formed. The deprotonation of the aminyl radical by TEOA itself leads to a rearrangement into a carbon centered radical displaying a significant reductive power. The iminium species further degrades into hydroxy ethanal and secondary amine by hydrolysis of the iminium in aqueous media. Therefore, triethanolamine degradation maybe do not form ethanol directly. Factually, if triethanolamine act as the sacrificial donors, many references have reported that the C_1 and C_2 products are indeed derived from the CO_2RR rather than the other carbon sources (please refer to: Energy Environ. Sci. 2022, 15, 225-233; J. Am. Chem. Soc. 2021, 143, 2984-2993; J. Am. Chem. Soc. 2020, 142, 2413-2428).

At the stage of laboratory research where the mechanism of photocatalytic CO_2 reduction is still unclear, the rational use of sacrificial agent donor helps researcher to bypass the oxidized half reaction, and focus more on the half-reaction of CO_2 reduction. Therefore, although triethanolamine can produce some unnecessary oxidation products in photosystem, it is still used to reveal the symmetry-forbidden coupling mechanism of C_1 intermediates on diatomic sites during CO_2RR . We sincerely hope that these above explanations can avoid the possible misunderstanding mentioned by this Reviewer.

Accordingly, the relative part (page 10) of analyses of CO_2 photoreduction activity was revised as: “ CO_2 reduction over various monomers and diatomic COFs was conducted using a liquid-phase photoreaction system, **with triethanolamine assistance as sacrificial agent upon oxidized process (Supplementary Fig. 17)**. The ZnPor monomer does not exhibit any CO_2RR activity (Fig. 3a). However, the production rate of H_2 , CO , and formate is higher for CuN_3 than for RuN_3 , indicating that the chelated copper centers can act as more efficient active sites for CO_2RR .”

Supplementary Figure 17. Mechanism of triethanolamine degradation as sacrificial reagent in photocatalytic system. Once interacting with the excited ZnPor photosensitive centers, a positively charged aminyl radical is formed. The deprotonation of the aminyl radical by TEOA itself leads to a rearrangement into a carbon centered radical displaying a significant reductive power. The iminium species further degrades into hydroxy ethanal and secondary amine by hydrolysis of the iminium in aqueous media.

7. In experiment, the authors used acetonitrile and why acetonitrile is used for solvent? If simple water is used for solvent, what product is obtained?

Re: Thanks a lot for this comment

A mixing solution (12 mL acetonitrile and 2 mL deionized water) are used to increase the CO₂ solubility in this work, which is a general strategy in CO₂ reduction (refer to: Energy Environ. Sci. 2022, 15, 225-233; Adv.Mater.2019, 31, 1900709; Angew. Chem. Int. Ed.2019,58, 5226 –5231). Furthermore, when optimizing the CO₂ reduction reaction conditions, we have tested the products of these diatomic COFs at pure water phase rather than mixing solution with the same condition. ZnPor-RuCuDAC show the higher acetate production rate (22.6 μmol g⁻¹ h⁻¹) as well as the lower formate (2.2 μmol g⁻¹ h⁻¹) and CO (1.8 μmol g⁻¹ h⁻¹) photoactivities; ZnPor-Cu₂DAC exhibit the higher CO evolution rate (17.2 μmol g⁻¹ h⁻¹) as well as the lower formate (2.6 μmol g⁻¹ h⁻¹) and acetate (3.2 μmol g⁻¹ h⁻¹) photoactivities; ZnPor-Ru₂DAC give the better CO (15.3 μmol g⁻¹ h⁻¹) photoactivity than the formate (1.5 μmol g⁻¹ h⁻¹) and acetate (1.2 μmol g⁻¹ h⁻¹) photoactivities. Notably, ZnPor-RuCuDAC exhibits the highest acetate selectivity in aqueous phase, while ZnPor-Cu₂DAC and ZnPor-Ru₂DAC presents the most superior CO selectivity, indicating that the active-site-type variations in the COF skeleton can influence the product photoactivity.

However, because of the inferior CO₂ solubility, the selectivity of these COFs toward acetate, CO and formate in the water phase is similar to the acetonitrile/water mixing solution, but the aqueous photoactivity is much lower than that of mixing solution. Therefore, **we hope to discuss the photoactivity and selective mechanism of these COFs toward CO₂ photoreduction in acetonitrile/water mixing solution at the revision if this Reviewer agree.**

8. Change of oxidation number of Ru and Cu seem to be important to determine the activity and so please measure the XPS for analysis of oxidation state of Ru and Cu before and after reaction.

Re: We appreciate the referee bringing this important issue to our attention.

Before and after 40 h photoreaction, XPS spectra of these diatomic COFs are added as Supplementary Fig. 13 and Supplementary Fig. 24 in supporting information. The recovered diatomic catalysts after the four consecutive photoreaction cycles exhibit XPS profiles very similar to that of their original one. The survey XPS (Supplementary Fig. 24a) spectra of the recovered COFs product after photocatalytic reaction indicates that Zn, Ru, Cu, C and N still coexist in the COFs' skeleton. The high-resolution recovered Zn2p, Cu2p and Ru 3p XPS (Supplementary Fig. 24b–d) spectra has Zn(II), Ru(II) and Cu(II) binding energy peaks, respectively, which are the same to that of diatomic COFs without long-time photoreaction. These results indicate that the element composition and oxidation states of ZnPor-RuCuDAC, ZnPor-Ru₂DAC and ZnPor-Cu₂DAC are unchanged during the long-term photoreaction process, and thus has very high structural stability and no significant destruction.

Accordingly, the relative part (page 11, 2nd paragraph) was revised as: “In addition to the high acetate selectivity of ZnPor-RuCuDAC, a stable photoactivity was also retained (Fig. 3f and Supplementary Fig. 20–22), and no obvious decay was observed during four consecutive runs. Moreover, **the diatomic catalyst and its recovered composite present quite similar DRS (Supplementary Fig. 23a), PXRD (Supplementary Fig. 23b) and XPS spectra (Supplementary Fig. 24), suggesting that crystallinity, element composition and oxidation state are still maintained after 40 h reaction.**”

9. Supplementary Figure 6, pore size distribution is shown, and I think wall thickness of COF can be estimated from the observed pore size and this is required for analysis of pore structure

of the obtained COF. What pore is corresponding to 5 nm which is weak peak in pore size distribution measurement?

Re: We thank the reviewer for professional comment.

The Barret-Joyner-Halenda (BJH) pore size distribution plots are determined from the desorption branches. ZnPor-RuCuDAC display two peaks centered at 1.66 and 4.32 nm, ZnPor-Ru₂DAC show two major peaks at 1.68 and 4.46 nm, and ZnPor-Cu₂DAC exhibit two peaks centered at 1.65 and 4.17 nm. The pore size of lower region (~1.66 nm) is the capillary condensation of N₂ molecules among the square-like pores within the COFs, while the pore size of high one can be ascribed to N₂ capillary condensation at the interlayer of COF nanosheet (referring to: J. Am. Chem. Soc. 2017, 139, 8705-8709; J. Am. Chem. Soc. 2019, 141, 17431-17440).

Accordingly, the relative part in the supporting information was revised as: “The Brunauer–Emmett–Teller (BET) surface areas were estimated as 902.3 (ZnPor-RuCuDAC COF), 867.3 (ZnPor-Ru₂DAC COF), and 826.3 (ZnPor-Cu₂DAC COF) m² g⁻¹, with pore volumes of 0.36, 0.31, and 0.25 cm³ g⁻¹, respectively. By the desorption branches, the Barret-Joyner-Halenda (BJH) pore size distribution plots are obtained. ZnPor-RuCuDAC display two peaks centered at 1.66 and 4.32 nm, ZnPor-Ru₂DAC show two major peaks at 1.68 and 4.46 nm, and ZnPor-Cu₂DAC exhibit two peaks centered at 1.65 and 4.17 nm. The pore size of lower region is the capillary condensation of N₂ molecules among the square-like pores within the COFs, while the pore size of high one can be ascribed to N₂ capillary condensation at the interlayer of COF nanosheet.”

10. Supplementary Figure 12, The authors estimated band gap from UV-VIS., but it seems estimated band gap is overestimated and should estimate from straight line in more wide area. In this case, the estimated narrow band gap should be discussed.

Re: Thanks a lot for this useful suggestion!

As seen in revision, after evaluating from the straight line, ZnPor-RuCuDAC, ZnPor-Ru₂DAC and ZnPor-Cu₂DAC can be obtained a narrow band gap with gap energy of 0.61 eV, 0.48 eV and 0.53 eV, respectively; a broad band gap can also be calculated for ZnPor-RuCuDAC (1.75 eV), ZnPor-Ru₂DAC (1.63 eV) and ZnPor-Cu₂DAC (1.66 eV). Since these

synthesized metalloporphyrin COFs have the claret color and hold with ca. 720 nm absorption edge from ultraviolet-visible diffuse reflection spectra, it can be conjectured that they are an intrinsic semiconductor with broad bandgap. If the bandgap of ZnPor-RuCuDAC, ZnPor-Ru₂DAC and ZnPor-Cu₂DAC is 0.61, 0.48, 0.53 eV, respectively, they would be a black color with characterization of fully absorbed light ability, which is obviously unscientific for the present metalloporphyrin materials. Many references have reported that the porphyrin-based COFs or polymers possess a bandgap energy from 1.5 eV to 2.1 eV (Nat. Commun. 2021, 12, 1354; J. Am. Chem. Soc. 2019, 141, 17431-17440; ACS Appl. Mater. Inter. 2019, 2, 5665-5676).

Correspondingly, the relative part in the supporting information was revised as following:

Supplementary Figure 15. (a) Ultraviolet-visible diffuse reflection spectra of the COFs. Obtained bandgaps by plotting $(\alpha h\nu)^2$ versus $h\nu$ for the (b) ZnPor-RuCuDAC, (c) ZnPor-Ru₂DAC and (d) ZnPor-Cu₂DAC, respectively. From Kubelka–Munk method, a broad band gap can be calculated for ZnPor-RuCuDAC (1.75 eV), ZnPor-Ru₂DAC (1.63 eV) and ZnPor-Cu₂DAC (1.66 eV), respectively; A narrow band gap can also be obtained for ZnPor-RuCuDAC

(0.61 eV), ZnPor-Ru₂DAC (0.48 eV) and ZnPor-Cu₂DAC (0.53 eV). Since these synthesized metalloporphyrin COFs have the claret color with ca. 720 nm absorption edge, it can be concluded that they are an intrinsic semiconductor with broad bandgap.

REVIEWERS' COMMENTS

Reviewer #1 (Remarks to the Author):

The raised comments have been well addressed and this revised version can be accepted for publication as it is.

Reviewer #2 (Remarks to the Author):

Thanks to the authors for their answers and supplements, but there is still one problem that has not been resolved.

It can be seen from Fig.1h-g that the distances between adjacent Zn, adjacent Cu and adjacent Ru are all ~ 1.67 nm. This result was also confirmed by Supplementary Figure 10, which was later added. However, it is obvious from the simulated structure in Fig. 1c and Supplementary Figure 4b given by the authors that the distances between adjacent Zn, adjacent Cu and adjacent Ru should be 2.33 nm (along the diagonal) or 3.34 nm (along the axis a or b). Therefore, we still believe that the constructed COF skeleton structure is inconsistent with the most intuitive electron microscope experimental results, no matter the previous data or the supplementary data.

Reviewer #3 (Remarks to the Author):

Revision is reasonably done and this version of manuscript could be accepted for publication.

Point-by-point responses to the reviewers' comments

- **Reviewer #1 (Remarks to the Author):**

The raised comments have been well addressed and this revised version can be accepted for publication as it is.

Re: Thanks a lot for your positive comments and support the publication of our manuscript.

- **Reviewer #2 (Remarks to the Author):**

Thanks to the authors for their answers and supplements, but there is still one problem that has not been resolved.

It can be seen from Fig. 1h-g that the distances between adjacent Zn, adjacent Cu and adjacent Ru are all ~ 1.67 nm. This result was also confirmed by Supplementary Figure 10, which was later added. However, it is obvious from the simulated structure in Fig. 1c and Supplementary Figure 4b given by the authors that the distances between adjacent Zn, adjacent Cu and adjacent Ru should be 2.33 nm (along the diagonal) or 3.34 nm (along the axis a or b). Therefore, we still believe that the constructed COF skeleton structure is inconsistent with the most intuitive electron microscope experimental results, no matter the previous data or the supplementary data.

Re: Many thanks for your valuable and professional comments.

We are sorry for our carelessness in the construction of the stacking model. To correctly understand the real COF skeleton structure, we have constructed another slipped ABCD-staggered stacking models in the revision with carefully referring literatures (J. Am. Chem. Soc. **2018**, 140, 16124-16133; Chem. Sci. 2021, 12, 6280–6286; ACS Appl. Mater. Interfaces 2021, 13, 25, 29471–29481), in addition to three types of possible 2D structures for ZnPor-RuCuDAC (AA-eclipsed, AB-staggered and slipped ABC-staggered). As seen in Fig. R1 (for review only), the distance of the adjacent Zn, adjacent Cu, and adjacent Ru are ca. 1.67 (along x/y axis) or 2.36 nm (along the diagonal) with ABCD stacking, while the spatial distance of the adjacent Zn, adjacent Cu, and adjacent Ru are ca. 2.36 (along the diagonal) or 3.34 nm (along x/y axis) according to the AB-staggered stacking models (Supplementary Figure 5b). **It should be noted that some Ru and Cu atoms are overlaid in Fig R1b due to the staggered four-atom layers, but they can be clearly observed in the three dimensions (Fig R1c).** Moreover, to

avoid misunderstanding because of the abundant atom overlap and better observe the spatial position of Zn, Cu, and Ru in the ABCD-staggered stacking model, the amplified CPK displaying-style structure (used in Fig. 1g) of ZnPor-RuCuDAC are shown in Fig. R2 for this reviewer consideration.

The simulated powder X-ray diffraction (PXRD) analyses reveal that the ABCD-staggered (A layer: $x = 0, y = 0$; B layer: $x = 1, y = 1$; C layer: $x = 1, y = 0$; D layer: $x = 0, y = 1$) stacking has the most intense peak at $\sim 4.41^\circ$ due to the (200) reflection plane, following with other minor peaks at 3.06° and 9.57° that originate from the (202) and (211) reflection planes, respectively, similar to the AB stacking models. The parameters of individual building compartments from AC-ADF-STEM (Fig. 1g) are measured with $a = b = 1.67 \pm 0.1$ nm ($\alpha = 90 \pm 0.3^\circ$) in ZnPor-RuCuDAC COF. The 3D electron diffraction tomography of DPC images (Fig. 1h–j) reveals the spatial distances between adjacent Zn, Cu, and Ru are ca. 1.67 nm, which is equably atomic cross-distribution on the ZnPor-RuCuDAC substrate. These results confirm that the COF photocatalysts possibly present the ABCD-staggered stacking structure rather than the simple AB-staggered configuration, which is evidenced by the negligible difference obtained in the Pawley refinement results.

Further, the density functional tight binding method with Lennard–Jones function was utilized to evaluate crystal stacking energies of ZnPor-RuCuDAC quantitatively (Supplementary Table 5). Evidently, the AA-eclipsed, AB-staggered, slipped ABC-1, slipped ABC-2, slipped ABC-3 and slipped ABCD stacking exhibit total per-layer crystal stacking energies of 57.82, 134.53, 97.25, 97.25, 118.79 and 193.95 kcal mol⁻¹, respectively, indicating that ABCD-staggered stacking is more favorable than the other five isomeric structures. according to the nonlocal density functional theory, these COFs with AA-eclipsed, AB-staggered, slipped ABC-1, slipped ABC-2, slipped ABC-3, and slipped ABCD stacking models show pore size distributions (Supplementary Fig. 4b–f) centered at 3.41, 1.67, 1.25, 1.25, 0.93 and 1.67 nm, respectively.

Therefore, taking these above results into consideration, we think the ABCD-staggered stacking model of these COF photocatalysts possibly presents its real structure better than the other models. Because of the cross-distribution of Zn, Ru, and Cu on the ZnPor-RuCuDAC nanosheet (~ 7 nm thickness), the spatial distances among adjacent Zn, adjacent Ru and adjacent Cu are ca. 1.67 nm as observed by AC-ADF-STEM. We are especially very grateful for this Reviewer's useful and friendly reminder on this issue and will focus on it to avoid such worries in our future research.

Fig. R1 Topological representation of ZnPor-RuCuDAC with the slipped ABCD-staggered stacking models viewed along the (a) x-axis and (b) z-axis as well as (c) its three dimensions (A layer: gray color; B-layer: magenta color; C-layer: yellow color; D-layer: green color). (for review only)

Fig. R2 The amplified CPK structure (inserted at Fig. 1g) of ZnPor-RuCuDAC with the slipped ABCD-staggered stacking viewed along the x-axis. (for review only)

According to this comment, the manuscript (From Page 6, last paragraph to Page 7, the first paragraph) is revised as: "...The parameters of individual building compartments are measured with $a = b = 1.67 \pm 0.1$ nm ($\alpha = 90 \pm 0.3^\circ$) in ZnPor-RuCuDAC COF, in good agreement with the ABCD-staggered pore size of simulated structure (inserted at Fig. 1g and Supplementary Fig. 4f,g). The 3D electron diffraction tomography of DPC images (Fig. 1h–j) unveil **the spatial distances among adjacent Zn, Cu and Ru are ca. 1.67 nm over** the ZnPor-RuCuDAC substrate, where the Zn and Ru/Cu atoms show a 90° phase difference on the spatial location (Supplementary Fig. 10). **These results confirm that the COFs possibly present ABCD-staggered stacking structure.**"

The related analysis of structure and unit cell parameter of these COFs (Page S25–S27) in supplementary information is revised as: "In order to elucidate the structures of these COFs and unit cell parameters, **four types** of possible 2D structures were generated for ZnPor-RuCuDAC: AA-eclipsed (Supplementary Fig. 4a), AB-staggered (Supplementary Fig. 4b), slipped ABC-staggered (Supplementary Fig. 4c–e) and slipped ABCD-staggered (Supplementary Fig. 4f,g) stacking models. The simulated powder X-ray diffraction (PXRD) analyses (Supplementary Fig. 4h and Supplementary Table 2–4) reveal that the AA-eclipsed (**A layer: $x = 0, y = 0$**) stacking shows the first intense peak at a low angle of 2.46° , corresponding to the (110) reflection; also, it shows minor peaks at 3.39° and 4.92° for the (202) and (004) reflection planes, respectively. For the AB-staggered (**A layer: $x = 0, y = 0$; B layer: $x = 1, y = 1$**) stacking, the most intense peak corresponding to the (200) reflection plane appears at $\sim 4.41^\circ$, along with other minor peaks at 3.06° and 9.57° that originate from the (202) and (211) reflection planes, respectively. The slipped ABC-staggered stacking exhibits the same PXRD peaks at $2.46^\circ, 3.39^\circ, 4.81^\circ, 6.62^\circ, 7.08^\circ, 7.89^\circ, 13.25^\circ,$ and 18.49° , corresponding to the (200), (202), (004), (210), (212), (014), (420), and (031) planes, respectively. Both the ABC-1 (**A layer: $x = 0, y = 0$; B layer: $x = 1, y = 1$; C layer: $x = 0.5, y = 0$**) and ABC-2 (**A layer: $x = 0, y = 0$; B layer: $x = 1, y = 1$; C layer: $x = 0, y = 0.5$**) stacking models show the most intense peak at 4.81° , whereas the slipped ABC-3 (**A layer: $x = 0, y = 0$; B layer: $x = 1, y = 1$; C layer: $x = 0.5, y = 0.5$**) model shows the strongest peak intensity at 3.39° . The simulated diffraction peaks of ABCD (**A layer: $x = 0, y = 0$; B layer: $x = 1, y = 1$; C layer: $x = 1, y = 0$; D layer: $x = 0, y = 1$**) model are similar to that of AB stacking. However, it should be noted that the spatial distance of the adjacent Zn, Cu and Ru are ca. 2.36 (along the diagonal) or 3.34 nm (along x/y

axis) according to the AB-staggered stacking models, while the distance of the adjacent Zn, Cu and Ru are ca. 1.67 (along x/y axis) or 2.36 nm (along the diagonal) with ABCD stacking.

Further, the density functional tight binding method with Lennard–Jones function was utilized to evaluate crystal stacking energies of ZnPor-RuCuDAC quantitatively (Supplementary Table 5). Evidently, the AA-eclipsed, AB-staggered, slipped ABC-1, slipped ABC-2, slipped ABC-3 and slipped ABCD stacking exhibit total per-layer crystal stacking energies of 57.82, 134.53, 97.25, 97.25, 118.79 and 193.95 kcal mol⁻¹, respectively, indicating that ABCD-staggered stacking is more favorable than the other five isomeric structures. The experimental PXRD profiles of ZnPor-RuCuDAC, ZnPor-Ru₂DAC, and ZnPor-Cu₂DAC exhibit three main diffraction peaks (Fig. 1b–d, experimental profiles in black, Pawley-refined profiles in red, predicted profiles in magenta, and differences in blue) at 3.06°, 4.42°, and 9.57°, which are consistent with those obtained in the simulated diffraction pattern of AB- and ABCD-staggered stacking model. This result suggests that these diatomic COFs adopt an AB- or ABCD-stacked structure. Moreover, according to the nonlocal density functional theory, these COFs with AA-eclipsed, AB-staggered, slipped ABC-1, slipped ABC-2, slipped ABC-3 and slipped ABCD stacking models show pore size distributions (Supplementary Fig. 4b–f) centered at 3.41, 1.67, 1.25, 1.25, 0.93 and 1.67 nm, respectively. However, the parameters of individual building compartments from AC-ADF-STEM (Fig. 1g) are measured with $a = b = 1.67 \pm 0.1$ nm ($\alpha = 90 \pm 0.3^\circ$) in ZnPor-RuCuDAC COF. The 3D electron diffraction tomography of DPC images (Fig. 1h–j) reveal the spatial distances between adjacent Zn, Cu and Ru are ca. 1.67 nm, which is equally atomic cross-distribution on the ZnPor-RuCuDAC substrate. These results confirm that the COF photocatalysts possibly present ABCD-staggered stacking structure rather than the simple AB-staggered configuration, which is evidenced by the negligible difference obtained in the Pawley refinement results.”

Fig. 1 Selective reaction mechanism and microstructural characterization. **a.** Expected photocatalytic CO₂ reduction mechanism of C₁ and C₂₊ product formation using homo- and hetero-diatomous catalytic sites, respectively. **b–d.** Pawley-refined PXRD of (b) ZnPor-RuCuDAC, (c) ZnPor-Ru₂DAC, and (d) ZnPor-Cu₂DAC. **e–j.** STEM analyses of ZnPor-RuCuDAC. **e,f:** (e) Filtered AC-BF-STEM images and (f) corresponding color-coded pore channel map. **g–j:** (g) DPC image using AC-ADF-STEM and (h–j) corresponding 3D electron diffraction tomography of DPC images. All microstructural analyses confirm the ABCD-staggered stacking structure of ZnPor-RuCuDAC COF.

Supplementary Figure 5. Topological representation of ZnPor-RuCuDAC. (a) AA-eclipsed, (b) AB-staggered, and (c–e) slipped ABC-staggered stacking models viewed along the z-axis (top) and x-axis (bottom). The slipped ABCD-staggered stacking models (f) viewed along the z-axis (top) and x-axis (bottom) as well as (g) its three dimensions. (h) Simulated PXRD patterns.

Supplementary Table 5. Unit cell parameters and crystal stacking energies of different stacking model for ZnPor-RuCuDAC COF.

Stacking model	AA	AB	ABC-1	ABC-2	ABC-3	ABCD
$a = b$ (Å)	38.25	40.57	41.31	41.31	42.57	40.63
c (Å)	5.53	3.81	4.68	4.68	4.03	5.76
$\alpha = \beta = \gamma$ (degree)	90	90	90	90	90	90
Space group	P2/M	P21/M	PMM2	PMM2	PCA21	P21/C
Formula	C ₂₅₆ N ₆₄ H ₁₅₂ Zn ₂ Ru ₄ Cu ₄	C ₂₅₆ N ₆₄ H ₁₅₂ Zn ₂ Ru ₄ Cu ₄	C ₃₈₄ N ₉₆ H ₂₂₈ Zn ₃ Ru ₆ Cu ₆	C ₃₈₄ N ₉₆ H ₂₂₈ Zn ₃ Ru ₆ Cu ₆	C ₃₈₄ N ₉₆ H ₂₂₈ Zn ₃ Ru ₆ Cu ₆	C ₅₁₂ N ₁₂₈ H ₃₀₄ Zn ₄ Ru ₈ Cu ₈
Total crystal stacking energy per layer (kcal mol ⁻¹)	57.82	134.53	97.25	97.25	118.79	193.95
Cohesive bulk energy per layer (kcal mol ⁻¹)	-57.82	-134.53	-97.25	-97.25	-118.79	-163.95

● **Reviewer #3 (Remarks to the Author):**

Revision is reasonably done and this version of manuscript could be accepted for publication.

Re: We thank the reviewer for these very positive comments and support the publication of our work.